# A Supportive Role of Mesenchymal Stem Cells on Insulin-Producing Langerhans Islets with a Specific Emphasis on The Secretome

**DOI:** 10.3390/biomedicines11092558

**Published:** 2023-09-18

**Authors:** Ronit Vogt Sionov, Ronit Ahdut-HaCohen

**Affiliations:** 1The Institute of Biomedical and Oral Research (IBOR), Faculty of Dental Medicine, The Hebrew University of Jerusalem, Jerusalem 9112102, Israel; 2Department of Medical Neurobiology, Institute of Medical Research, Hadassah Medical School, The Hebrew University of Jerusalem, Jerusalem 9112102, Israel; ronit.ahdut@mail.huji.ac.il; 3Department of Science, The David Yellin Academic College of Education, Jerusalem 9103501, Israel

**Keywords:** β-cells, growth factors, insulin, Langerhans’ islets, mesenchymal stem cells

## Abstract

Type 1 Diabetes (T1D) is a chronic autoimmune disease characterized by a gradual destruction of insulin-producing β-cells in the endocrine pancreas due to innate and specific immune responses, leading to impaired glucose homeostasis. T1D patients usually require regular insulin injections after meals to maintain normal serum glucose levels. In severe cases, pancreas or Langerhans islet transplantation can assist in reaching a sufficient β-mass to normalize glucose homeostasis. The latter procedure is limited because of low donor availability, high islet loss, and immune rejection. There is still a need to develop new technologies to improve islet survival and implantation and to keep the islets functional. Mesenchymal stem cells (MSCs) are multipotent non-hematopoietic progenitor cells with high plasticity that can support human pancreatic islet function both in vitro and in vivo and islet co-transplantation with MSCs is more effective than islet transplantation alone in attenuating diabetes progression. The beneficial effect of MSCs on islet function is due to a combined effect on angiogenesis, suppression of immune responses, and secretion of growth factors essential for islet survival and function. In this review, various aspects of MSCs related to islet function and diabetes are described.

## 1. Introduction

Type 1 diabetes (T1D) or juvenile diabetes is a chronic autoimmune disease in which insulin-producing β-cells in the endocrine pancreas are gradually destroyed by immune cells, eventually leading to insufficient insulin production and uncontrollably fluctuating serum glucose levels [1,2,3,4]. Type 2 diabetes (T2D) is a metabolic disease where peripheral tissues, such as muscle and liver, have developed resistance to insulin signaling, reducing the ability of the tissues to take up glucose, eventually leading to hyperglycemia [4,5,6]. T2D can also develop when the β-cell mass decreases as a result of the cytotoxic effects of chronic hyperglycemia, chronic low-grade inflammation, excessive reactive oxygen species production, endoplasmic reticulum (ER) stress, and islet amyloid polypeptide deposition [7,8,9,10]. Glucose tolerance is often reduced in the elderly due to a combined effect of peripheral insulin resistance and impaired insulin secretion [11,12,13,14].

T1D is characterized by chronic inflammation and immune cell infiltration of the islets in a process termed insulitis [15,16,17,18,19,20,21]. Both hypoglycemia and hyperglycemia lead to health complications [22,23]. Hyperglycemia leads to macrovascular and microvascular complications, such as retinopathy, nephropathy, neuropathy, and cardiovascular diseases [24,25]. Chronic hyperglycemia also leads to alterations in islet cytoarchitecture with α-cell hyperplasia, β-cell transdifferentiation into glucagon-secreting cells, and deregulated hormone secretion [26]. T2D patients exhibit elevated glucagon secretion [26,27], and T1D patients secrete more glucagon during mixed-meal stimulation [28,29]. The elevated glucagon levels in T2D individuals may be due to α-cell resistance to insulin and somatostatin, whose function is to reduce glucagon secretion [26,30,31]. Thus, a vicious cycle is generated.

The destruction of β-cells is mediated by cytotoxic CD8^+^ T lymphocytes that mistakenly recognize β-cells as foreign bodies, but other immune cells also contribute to this process, including B lymphocytes that produce autoantibodies, and macrophages, dendritic cells, and neutrophils, which produce cytokines, chemokines, reactive oxygen and nitrogen species, and other bioactive molecules, and act as antigen-presenting cells [1,3,15,16,17,18,19,21,32,33,34,35,36,37]. Besides modulating immune responses, the combined production of inflammatory cytokines such as TNFα, IL-1β, and IFNγ, is detrimental to β-cells [35,38,39,40,41,42,43]. These cytokines cause mitochondrial dysfunction and endoplasmic reticulum stress, induce the expression of pro-apoptotic molecules, and activate apoptotic pathways in β-cells [3,43,44,45,46,47]. 

To maintain adequate blood glucose levels, T1D patients need exogenous insulin administration in the form of subcutaneous injections. Alternative treatments include islet cell transplantation or whole pancreas transplantation, which requires subsequent immunosuppressive therapy [48,49,50,51,52,53]. Up to 80% of the transplanted islets are lost before becoming integrated into tissue due to acute inflammatory responses and release of the pro-inflammatory cytokines IL-1β, TNFα, and IFNγ [9,48,54,55,56,57]. The success of islet transplantation is also challenged by allo-immune graft rejection and recurrence of autoimmunity [58,59,60]. Moreover, the supply of donor tissues is limited.

Mesenchymal stem cells (MSCs) are multipotent non-hematopoietic progenitor cells found in various tissues, including the bone marrow, adipose tissue, liver, and umbilical cord blood. They can differentiate into various cell types, including osteocytes, adipocytes, chondrocytes, endothelial cells, and myocytes [61]. Their low immunogenicity, together with immunosuppressive properties, has made MSCs a promising therapeutic tool for various autoimmune diseases, including T1D [48]. Several studies have shown that MSCs, by virtue of their immunomodulatory and pro-angiogenetic effects, can attenuate immune responses and enhance islet engraftment following transplantation [62,63]. This review focuses on the beneficial effects of MSCs on β-cell function, with a specific emphasis on the secretome. A brief introduction to insulin-producing β-cells and the hazardous effects of pro-inflammatory cytokines on β-cells proceeds the discussion on the different aspects of MSCs involved in preserving β-cell function.

## 2. Insulin-Producing β-Cells

Insulin-producing β-cells are the major cell type of Langerhans islets (around 60% of the islet cells), intermixed with other cell types, including glucagon-producing α-cells (around 30% of the islet cells), somatostatin-producing δ-cells (less than 10% of islet cells), pancreatic polypeptide-producing γ or PP-cells (less than 5% of islet cells), ghrelin-producing ε-cells, supportive pericytes and contractile smooth muscle cells [64,65,66,67]. The endocrine pancreas is not a single organ, but it is rather composed of millions of islets scattered throughout the exocrine pancreas [64,65], although some clusters of small islets have been found in the human pancreas [65]. These smaller islets consist of more β-cells and have a higher insulin content than the large islets [68,69]. Concerted regulation of insulin secretion and glucagon secretion is important for maintaining glucose homeostasis [64]. 

### 2.1. Vascularization of the Islets

The islets are highly infiltrated by blood vessels, enabling immediate sensation of changes in serum glucose levels as well as direct and prompt secretion of insulin and glucagon into the bloodstream as needed. The blood vessels also deliver oxygen required for β-cell function and survival [70]. Approximately 10% of the blood flow in the pancreas is delivered to the pancreatic islets despite comprising only 1–2% of the tissue mass. The smaller islets are frequently found clustered around the microcapillary beds of the endocrine pancreas [65,70], compensating for their lack of intra-islet capillaries [66,71]. In contrast, the larger islets are supplied by up to three arterioles [66,70,71]. The intra-islet endothelial cells, which are attracted by β-cells through the secretion of vascular endothelial growth factor (VEGF)-A [72], enhance insulin secretion and stimulate β-cell proliferation [73], among others, through the production of basement membrane proteins, such as laminins [74,75]. The production of the angiogenetic factor angiopoietin-1 (ANG1) by β-cells stabilizes the blood vessels in the islets, which indirectly affects insulin secretion and glucose homeostasis [76]. Gan et al. [77] emphasized the importance of extracellular matrix proteins in β-cell function. This research group observed enriched insulin granule fusion in culture β-cells that have adhered to the extracellular matrix, which was dependent on β1 integrin receptor activation [77]. The importance of the basement membrane in supporting β-cell insulin secretion is further exemplified by the improved islet function and β-cell function observed when seeded on various extracellular matrix components or on tissue decellularized extracellular matrices, especially of the lung [55,78,79,80,81]. The incorporation of laminin and collagen IV into islet alginated microcapsules protected the islets from cytokine-mediated cell death [82].

### 2.2. Innervation of the Islets

Moreover, the islets are highly innervated, and their function is affected by signals delivered by neurotransmitters of both the sympathetic and parasympathetic nervous systems [83,84,85,86]. The brain perceives glucose levels both directly and indirectly, transducing signals to regulate islet function [85,87,88,89]. Particular attention has been paid to the inhibitory neurotransmitter γ-aminobutyric acid (GABA) [90,91]. Long-term exposure of α-cells to GABA resulted in reduced glucagon secretion and transdifferentiation into β-like cells [92,93]. Treatment of human islets with GABA resulted in decreased α-cell content with a concomitant increased β-cell content [92].

### 2.3. Cell Communication within the Islets

There is also continuous communication between the cells within the islets, with mutual modulation of the activity of neighboring cells [84,94,95,96,97]. This communication is mediated by paracrine factors and juxtacrine mechanisms involving conduction of electric waves through gap junctions formed by connexin-36 (Cx36) [71,94,96,98,99,100,101]. Insulin signals α-cells to reduce glucagon secretion via the insulin receptor and the GABA-GABA-A receptor system [31,102,103,104]. Vice versa, glucagon has an impact on β-cell function [26]. Prevention of glucagon signaling using a neutralizing antibody to the glucagon receptor promoted β-cell survival and increased insulin secretion [105]. β- and δ-cells are electrically coupled through gap junctions [95]. Glucose-mediated depolarization of β-cells leads to coupled δ-cell depolarization with consequent secretion of somatostatin from δ-cells and somatostatin-mediated inhibition of α-cell glucagon secretion [95]. Glucagon secretion from α-cells is also regulated by β-cells in a juxtacrine manner, where ephrin ligands on the β-cells interact with EphA receptors on α-cells, resulting in reduced glucagon secretion [106,107]. The EphA-ephrin A system is also involved in β-cell to β-cell communication and regulates insulin secretion [108,109]. In this case, EphA receptor phosphorylation provides forward inward signals that inhibit insulin secretion, while glucose stimulation leads to dephosphorylation of EphA, allowing ephrin A reverse signaling that enhances insulin secretion [108,109]. The β-cell to β-cell communication ensures low insulin secretion during starvation while enhancing glucose-stimulated insulin secretion [108].

## 3. Destruction of β-Cells by Cytokines

The pro-inflammatory cytokines IFNγ, IL-1β, and TNFα secreted by various immune cells involved in islet inflammation (insulitis), induce apoptosis of β-cells, and together with other immune cell reactions, including FasL, perforin, granzyme, and the nitric oxide radical (NO·), contribute to the inflammatory-induced reduction in β cell mass [3,40,44,45,46,110,111,112,113,114,115,116,117,118]. The pro-inflammatory cytokines also induce chemokine production by the islet β-cells, which further exaggerates inflammation by attracting additional immune cells to the already inflamed site [115,119,120]. Each cytokine can act on its own, but the combination of two or three of them leads to large alterations in gene expression that ultimately impair β-cell survival and function [40,46,112,113,114,121,122,123,124,125]. Non-obese diabetic (NOD) mice lacking TNFα receptor 1 (TNFR1 or TNFRp55) or IL-1 receptor showed delayed onset of diabetes [117,126]. Blocking TNFα with Etanercept, a human tumor necrosis factor receptor (TNFR) p75 Fc fusion protein, resulted in lower A1C levels and increased insulin production in children with early-onset T1D, suggesting that this treatment may preserve β-cell function [127]. Quattrin et al. [128] used a neutralizing antibody to TNFα (Golimumab) in a clinical trial in children and young adults with early-onset T1D, which resulted in improved endogenous insulin secretion, but all patients still required exogenous insulin.

### 3.1. Signal Transduction Pathways Induced in Islets and β-Cells by Pro-Inflammatory Cytokines

IL-1β, IFNγ, and TNFα induce different as well as parallel signal transduction pathways, which act in concert to induce β-cell apoptosis. While IL-1β and TNFα activate the NFκB signaling pathway, IFNγ acts primarily through Janus kinase (JAK)-mediated activation of the transcription factor STAT1 [125,129,130,131,132,133,134]. NFκB-mediated signaling is pro-apoptotic in β-cells, whereas it induces anti-apoptotic pathways in most other cell types [41,132,133,135,136]. Inhibition of NFκB signaling protected β-cells from cytokine-induced apoptosis and increased islet survival after transplantation [135,136,137,138]. The cytokine-induced activation of NFκB reduced *PDX1*, *NKX2-2*, *SLC2A2*, *MAFA*, *GLUT2*, and insulin (*INS1*) gene expression while increasing *c-MYC* expression in β-cells [139,140]. Thus, these cytokines can contribute to the dedifferentiation of β-cells [139]. TNFα and IL-1β also activate the p38 and JNK mitogen-activated protein kinases (MAPKs) in β-cells [129]. Excessive p38 and JNK activation by IL-1β has been associated with β-cell apoptosis [141,142]. 

### 3.2. Gene Expression Altered in Islets and β-Cells by Pro-Inflammatory Cytokines

Several transcriptome and microarray analyses have been performed to pinpoint the genes affected in islets or β-cells in response to the pro-inflammatory cytokines [39,45,114,140,143,144,145,146,147,148]. Cytokine-induced genes relevant to insulitis and β-cell damage included inducible nitric oxide synthase (iNOS) [39,114,140,143,149,150,151,152,153], caspase 1 [39,154], cyclooxygenase (COX)-2 [149,150], monocyte chemoattractant protein (MCP)-1/chemokine (C-C motif) ligand 2 (CCL2) [143,155] and other chemokines (e.g., CCL5, CCL3, CXCL9, CXCL10, CXCL11, IL-6, and IL-8) [114,143,146]. The IL-1β-mediated induction of COX2 was found to depend on nitric oxide production [149]. The pro-inflammatory cytokines were found to activate both the extrinsic and the intrinsic apoptotic pathways in β-cells [40,156,157]. Cottet et al. [156] observed that TNFα, but not IL-1β, activates caspase 8 in a β-cell line. Gunnet et al. [40] showed that exposure of β-cells or islets to the three cytokines IFNγ, IL-1β, and TNFα resulted in the dephosphorylation of Bad, activation of Bax-dependent mitochondrial stress, cleavage and activation of caspase 9 and caspase 3. The p53 upregulated modulator of apoptosis (PUMA) and the pro-apoptotic Bim were found to be upregulated in human islets and mouse β-cells after exposure to IL-1β/IFNγ or TNFα/IFNγ [157,158]. PUMA and Bim act upstream of Bax/Bak and induce the translocation of these proteins to mitochondria [159]. Silencing of PUMA or Bim partially protected β-cells from TNFα/IFNγ-induced apoptosis [157]. 

The pro-inflammatory cytokines downregulate the anti-apoptotic Mcl-1 in β-cells, thereby further increasing the susceptibility of the β-cells to the pro-apoptotic molecules of the intrinsic apoptotic pathway [160]. IL-1β increased iNOS expression, which is further enhanced by IFNγ in both rat and human islets [143,161]. In addition, IL-1β induces the expression of Death protein 5 (DP5)/Harakiri (Hrk) in rat islets [143] and rat INS-1 β-cell line [134], and IL-1β together with IFNγ induces ER stress in β-cells with phosphorylation of eukaryotic initiation factor 2α (eIF2α), induction of activating transcription factor 4 (ATF4) and upregulation of CCAAT/enhancer-binding protein (C/EBP) homologous protein (CHOP) [47,162,163]. Moreover, TNF superfamily member 10* (TNFSF10; TRAIL)* expression was increased in human β cells after exposure to IL-1β and IFNγ [45]. While TNFα increases the expression of the antiapoptotic X-linked inhibitor of apoptosis (XIAP), IFNγ represses this induction [125]. 

Nakayasu et al. [39] performed a comprehensive analysis of protein changes occurring after treatment of human islets with IL-1β and IFNγ. This study showed that cytokine treatment affected proteins related to NFκB signaling, cytokine-cytokine receptor interactions, apoptosis, antigen processing and presentation, and extracellular matrix [39]. Notable, IL-1β and IFNγ upregulated the expression of several interleukins (e.g., IL-11, IL-1α, IL-1β, and IL-32), chemokines (e.g., CCL5, CCL8, CCL13, CSF1, CXCL2,3,5,6,8,9,10,11, and CX3L1), various caspases (e.g., caspases 1, 4, 5, 7, 8, and 10), the receptor-interacting serine/threonine protein kinase 2 (RIPK2), the anti-apoptotic protein PUMA and the inducible nitric oxide synthase (iNOS) responsible for nitric oxide production [39]. The anti-apoptotic growth factors thrombospondin 1, connective tissue growth factor (CTGF), and osteopontin (SPP1) were downregulated by IL-1β and IFNγ [39]. Osteopontin protects β-cells from cytotoxic effects and prevents hyperglycemia [164]. Nakayasu et al. [39] further showed that IL-1β and IFNγ downregulated growth/differentiation factor 15 (GDF15, formerly known as macrophage inhibitory cytokine 1 [MIC-1]). Treating human islets with GDF15 prevented the apoptosis induced by IL-1β and IFNγ, and administration of GDF15 to non-obese diabetic (NOD) mice prevented the development of diabetes [39]. In a transcriptome analysis, Eizirik et al. [146] observed a significant downregulation of growth differentiation factor 10 (GDF10), fibroblast growth factor 17 (FGF17), and transforming growth factor β2 (TGFβ2) in human islets treated with IL-1β and IFNγ. 

RIPK2 (RIP2, CARDIAK) is involved in transmitting signals from nucleotide-binding oligomerization domain 1 (NOD1), NOD2, and Toll-like receptors (TLRs) to NFκB, resulting in the induction of cytokine production [165]. Caspase 1 is involved in the processing of pro-IL-1β and pro-IL-18 into mature inflammatory cytokines and was therefore initially named IL-1 beta converting enzyme [166,167,168]. Caspase 1, together with the apoptosis-associated speck-like protein containing a CARD (ASC) and the nucleotide-binding oligomerization domain, leucine-rich repeat, and pyrin domain-containing protein (NLRP) 3, forms the inflammasome, which is activated by several endogenous and exogenous stimuli (e.g., pathogen-associated molecular patterns (PAMPs) and damage-associated molecular patterns (DAMPs) leading to the activation of caspase 1 [167,168,169,170,171]. It means that it is not sufficient that pro-caspase-1 is transcribed, but it must also be activated. Caspase 4, which is also upregulated by cytokines in β-cells, is involved in the activation of caspase 1 [172]. Mitochondrial DNA from diabetic mice and reactive oxygen species (ROS) can activate caspase 1 [173,174,175]. RIPK2 has been shown to be an activator of pro-caspase-1 [176], resulting in the induction of neuronal cell death among others through caspase 1-mediated cleavage of Bid to truncated Bid (tBid) [177]. Overexpressing RIPK2 in MCF7 breast carcinoma cells resulted in apoptosis that was mediated through its caspase recruitment domain (CARD) [178]. The RIPK2-Caspase 1 signaling pathway is also involved in pyroptosis, a kind of lytic cell death caused by inflammation, also known as gasdermin-dependent cell death [169,179,180,181,182,183,184]. Caspase 1 cleaves gasdermin D to release a pore-forming domain that forms pores in the plasma membrane, leading to cell lysis [182]. NLRP3 deficiency prevented the development of T1D and improved glucose tolerance and insulin sensitivity in mice [175], which was associated with diminished T-cell activation, T helper 1 (Th1) differentiation, T cell chemokine expression, and pathogenic T cell migration to pancreatic islets [185]. Polymorphisms in the NLRP1 and NLRP3 genes have been associated with a predisposition to T1D [168,186,187].

Dad1, which is downregulated by the cytokines, regulates N-linked glycosylation, binds to the anti-apoptotic Mcl-1, and inhibits apoptosis [188,189]. Deletion of Dad1 in mice leads to aberrant embryonic morphology, impaired mesodermal development, and excessive apoptosis, ultimately resulting in lethality by embryonic day 10.5 [190,191,192]. These studies suggest an important role of Dad1 as a survival factor. Notably, Dad1 was upregulated in primary rat β-cells exposed to 10 mM glucose and 20 mM glucose compared to those exposed to 5 mM glucose [193].

The upregulation of A20 may be a mechanism to protect the β-cells from apoptosis [194,195,196]. Overexpression of A20 in islets increased the survival rate of allogeneic islet transplants by preventing NFκB signaling [197]. TLR signaling in immune cells might have both pro- and anti-diabetogenic effects affected by the gut microbiota [198,199,200]. TLR4 deficiency reduces macrophage infiltration into the islets [199]. TLR4 levels are upregulated in pancreatic islets of obese mice, and TLR4 knockout mice become less obese when fed with a high-fat diet [199]. TLR4 deficient β-cells isolated from mice fed with a high-fat diet showed improved glucose-stimulated insulin secretion and expressed lower mRNA levels of IL-6, TNFα, and MCP-1 [199]. Thus, upregulation of TLR4 on β-cells in response to fatty acids leads to increased cytokine and chemokine production, which promotes macrophage infiltration of the islets with resulting β-cell dysfunction [199]. Burrows et al. [198] observed that deletion of the TLR-associated Innate immune adaptor myeloid differentiation primary response gene 88 (MyD88) in NOD mice led to T1D development in germ-free, but not in germ-exposed, environments. They further observed that knocking out the TIR-domain containing adapter inducing IFNβ (TRIF) in the MyD88 knockout NOD mice led to T1D development under normal germ exposed conditions [198]. These observations suggest that TRIF, which acts downstream to TLR4, induces microbiota-induced tolerogenic pathways [198]. However, knocking down TLR2 in the MyD88 knockout NOD mice led to reduced T1D incidences in germ-free conditions, suggesting that TLR2 delivers pro-diabetic signals [198]. TLR2 knockout mice were less prone to streptozotocin-induced diabetes [200]. In an overexpressing study using 293 embryonic kidney epithelial cells, activation of TLR2 was found to induce apoptosis through activation of the MyD88-Fas-associated death domain protein (FADD)-caspase 8 and caspase 1 pathway [201]. Further studies are required to understand the contribution of cytokine-induced TLR2 expression to β-cell viability and function.

### 3.3. Cytokines and Growth Factors Promoting β-Cell Survival and Preventing Pro-Inflammatory Cytokine-Induced Apoptosis

The pro-apoptotic effect of the pro-inflammatory cytokines on β-cells can be antagonized by the anti-inflammatory cytokines IL-4, IL-6, IL-10, and IL-13, which activate Signal transducer and activator of transcription 3 (STAT3; IL-6 and IL-10) and STAT6 (IL-4 and IL-13) signal transduction pathways [43,202,203,204,205,206]. IL-4 promotes the production of protective regulatory Th2 cells [207,208]. IL-10, TGFβ, and IL-33 can prevent β-cell damage by suppressing the immune system and inducing immune tolerance [209]. Growth hormone protected β-cells from the deleterious effects of cytokines by activating STAT5 with a concomitant increase in the Bcl-xL/Bax ratio [210]. The cytokines also down-regulate the expression of the anti-apoptotic Mcl-1 [160]. Overexpression of Bcl-xL or Mcl-1 protects β-cells from cytokine-induced apoptosis [160,211]. Other factors that can protect β-cells from cytokine-induced cell death include islet neogenesis-associated protein (INGAP) and its active pentadecapeptide core [212], n-3 polyunsaturated fatty acids (n-3 PUFAs) [213], insulin [214], insulin-like growth factor 1 (IGF1) [215], IGF2 [216,217], hepatocyte growth factor [218], osteopontin [219], stromal cell-derived factor 1 (SDF-1) [220], and neutral ceramidase [221] (Table 1). The importance of IGF2 production from pancreatic mesenchymal cells in β-cell survival was demonstrated in a conditional IGF2 mouse model, where IGF2 deletion resulted in both acinar and β-cell hypoplasia [222]. Co-transplantation of islets with neural crest stem cells increased β-cell proliferation and improved islet function [223], which has been related to the secretion of nerve growth factor (NGF) [224,225]. Inhibition of NGF signaling increased basal insulin secretion but impaired glucose-stimulated insulin secretion [224].

Glucose at normal levels promotes the expansion and survival of β-cells [193,296,381,382,383], but it can also act as a stressor that induces β-cell dysfunction through glucotoxicity [384,385,386,387,388]. Glucose-mediated stimulation of β-cell growth and survival depends on the activation of the insulin receptor and insulin receptor substrate 2 [389]. Glucose increases the expression of prolactin (PRLR), growth hormone (GHR), cholecystokinin A (CCKAR), and glucose-dependent insulinotropic polypeptide (GIPR) receptors in primary rat β-cells [193]. 

A transcriptome analysis of genes altered following glucose treatment of a human β-cell line showed a rapid upregulation of the proconvertase PCSK1 involved in the proteolytic conversion of pro-insulin to insulin [390]. A similar upregulation of PCSK1 was observed in mouse islets exposed to glucose [384]. Transcriptome analysis of mouse islets exposed to high glucose showed upregulation of genes associated with enhanced respiration, ER stress, and oxidative stress [384]. Among the highly upregulated genes by high glucose is thioredoxin interacting protein (TXNIP; thioredoxin-binding protein 2 (TBP2)), which inhibits the antioxidant activity of thioredoxin (TRX), resulting in intracellular oxidative stress [391]. TXNIP is involved in the glucotoxic effects leading to β-cell death [387,392]. A proteomic analysis of glucose-treated human β-cells showed enrichment in proteins involved in translation, glycolysis, TCA metabolism, and insulin secretion [390]. The mTOR signal pathway was shown to be involved in the glucose-induced effects in human β-cells [390,393]. Bertolini et al. [227] observed that glucose increases the expression of activin B and its receptor ALK7 but downregulates activin A in mouse islets. This might be a feedback mechanism as activin B decreases glucose-stimulated Ca^2+^ influx through ALK7, while activin A increases the glucose-stimulated Ca^2+^ influx [227]. Glucose stimulation of mouse islets also leads to a transient induction of growth differentiation factor 5 (GDF5, also known as BMP14) and the transcription factor mesenchyme homeobox 2 (MEOX2) (unpublished data). Overexpressing of PDX1 in MIN6 β-cell line induced expression of both GDF5 and MEOX2 (unpublished data), suggesting that the expression of these genes is regulated by PDX1, which is a master regulator of β-cells [394,395,396]. GDF5 has been shown to form heterodimers with BMP2 and BMP4 [397], both of which modulate β-cell differentiation during embryonic development and regulate glucose-induced insulin secretion in adult islets (Table 1). The mesenchyme homeobox 2 (MEOX2) regulates vertebrate limb myogenesis [398,399] and is expressed in the vertebrate embryo in regions of epithelial–mesenchymal interactions [400]. MEOX2 expression has previously been shown to be expressed in MIN6 β-cells [401,402]. Further studies are required to understand the role of GDF5 and MEOX2 in β-cell survival and function.

## 4. Mesenchymal Stem Cells

Mesenchymal stem cells, also called multipotent stromal cells (MSCs), are adherent, spindle-shaped, fibroblast-like cells that can be isolated from various tissues, including the bone marrow, adipose tissue, and umbilical cord [403,404]. MSCs lack any markers of hematopoietic cells (e.g., CD34, CD45, CD19 and HLA-DR) and the endothelial marker CD31, but express CD105 (SH2 or endoglin), CD71, CD73, CD44, CD29, stem cell antigen-1, and CD90 (Thy-1) [405,406,407,408]. MSCs do not express the major histocompatibility complex II (MHC-II) or the co-stimulatory molecules B7-1, B7-2, CD40, and CD40L required for T cell activation, such that these cells cannot activate T lymphocytes, and rather most studies show that MSCs actually suppress T-cell proliferation and activity, and increase the proportion of T regulatory cells [409,410,411,412,413,414,415,416]. MSCs exhibit general immunosuppressive activities that are beneficial in the treatment of various autoimmune diseases [417,418,419,420]. MSCs can induce a T helper 1 (Th1) to T helper 2 (Th2) shift with reduced IFNγ secretion and increased IL-4 production [410]. MSCs suppress IFNγ secretion from IL-2-stimulated NK cells [410] and inhibit IL-15-induced NK cell proliferation and their production of IFNγ, TNFα and IL-10 [421]. MSCs modulate the activity and polarization of macrophages [422,423], dendritic cells [410,424,425,426,427,428,429], and neutrophils [430,431,432], thus contributing to the homeostasis of the inflammatory microenvironment. There is also a crosstalk between MSCs and immune cells with mutual regulation [420].

MSCs are characterized by high self-renewability and multipotency with the ability to differentiate into various cell lineages, including osteoblasts of the bone, myoblasts of the muscle, chondrocytes of the cartilage, and adipocytes of the adipose tissue [61,403,408,433,434,435]. There are several lines of evidence that MSCs are formed from the differentiation of perivascular pericytes [407,434,436,437,438]. Tissue-specific MSC functions have been suggested, where the local microenvironment may influence their plasticity [439].

Many protocols have been developed to differentiate MSCs into functional insulin-producing β-cells by exposing the cells to chemical and biological factors or by genetic manipulation introducing the PDX1 gene, which is a master regulator in pancreas organogenesis [232,437,440,441,442,443,444,445,446,447,448,449,450,451,452,453,454,455,456,457,458,459]. A common dominator for the different differentiation protocols is the sequential exposure of MSCs to different combinations of growth factors (e.g., EGF, bFGF, betacellulin, activin A, HGF, extendin-4, insulin) chemical compounds (e.g., nicotinamide), and B27 supplement (containing among others insulin, biotin, vitamin E, Vitamin A, selenium, putrescine, transferrin, catalase, superoxide dismutase, triodo-L-thyronine, linoleic and linolenic acids) for different time periods [441,442]. Nicotinamide enhances the differentiation of human pancreatic cells and promotes the expression of insulin, glucagon, and somatostatin [460]. It also induces MAF1 and insulin promoter activity in a rat β-cell line [461]. Nicotinamide protects β-cells from oxidative stress, by virtue of its antioxidant properties [462]. Gao et al. [231] used a five-step protocol to differentiate β-cells from MSCs. This protocol included an initial induction using the demethylation agent 5-aza-2′-deoxycytidine, followed by incubation in a low glucose medium. This was followed by serial incubations with activin A, all-trans retinoic acid (ATRA), and bFGF together with B27, insulin, transferrin, selenite, and nicotinamide. Scuteri et al. [463], however, observed that incubating rat MSCs with rat islets was sufficient to differentiate the MSCs into PDX1-expressing and insulin-secreting cells, suggesting that factors secreted by islets (e.g., insulin) can affect the phenotype of the interacting MSCs. Although the direct effect of insulin as a single differentiation factor on MSCs has not yet been documented, insulin has been shown to increase glucose uptake and GLUT4 translocation in MSCs [464]. The ability of MSCs to adhere to the islets [463] and to home to the islets following transplantation, where it improves β-cell function, suggests a mutual interaction between the two cell types. Similar to MSCs, human liver stem-like cells (HLSC) have been shown to generate insulin-producing 3D spheroid structures in vitro that could restore normoglycemia in streptozotocin-induced diabetic mice [465,466]. 

MSCs have been shown to improve the medical conditions of a variety of immune-mediated diseases, including graft rejection, graft-versus-host disease, rheumatoid arthritis, systemic lupus erythromatosis, Crohn’s disease, colitis, osteoarthritis, multiple sclerosis, experimental autoimmune encephalomyelitis (EAE), and psoriasis [467,468,469,470,471,472,473,474,475]. Moreover, MSCs can promote wound healing and tissue regeneration [404,476,477,478]. Human MSCs have been shown in various settings to have a beneficial role in diabetes [479,480,481]. There is accumulating evidence that MSCs have beneficial effects on insulin-producing β-cells and islet survival, and co-transplantation of islets with MSCs increases the survival of the islet grafts, which will be further discussed below. MSC transplantation has the advantage of being well tolerated by the patients without any apparent toxicity [482,483,484,485,486,487,488,489,490], although some occasionally adverse effects have been noted, such as gastrointestinal and skin disorders [480].

After transplantation of MSCs, these cells can home to injured tissues and promote tissue regeneration, among others, by differentiating into various cellular phenotypes, providing cytokines, chemokines, growth factors, and other bioactive factors, enhancing the proliferation of stem cells and progenitors of the tissue and suppressing immune responses [404,434,468,469,478,491]. Since many of the MSC functions are caused by secretory molecules, A. I. Caplan suggested renaming the cells to “medicinal signaling cells” [434]. 

Using luciferase-expressed MSCs, Lin et al. [492] observed that intra-arterially injected MSCs specifically engraft to sites of injury caused by local irradiation of mice. Similarly, Chapel et al. [493] observed that green fluorescence protein (GFP)-labeled MSCs home to injured tissues in a model of total body irradiation of macaques (*Macaca fascicularis*). By transplanting MSCs from male rats into female rats, Boumaza et al. [494] found that MSCs can also be found in the pancreas. DiR-labeled human umbilical cord-derived MSCs were found to accumulate in the lung, liver, spleen, and pancreas for up to 7 days after intravenous injection into streptozotocin-induced diabetic mice [495]. The homing of the MSCs to the pancreas is believed to have a supportive role in islet regeneration and survival. 

### 4.1. In Vitro Evidence for β-Cell Supporting Roles of MSCs

Cultivation of human islets in vitro leads to loss of function, dedifferentiation, senescence, apoptosis, and necrosis [63,78,340,379,496,497,498]. The isolation process also reduces the number of viable islets. Single-cell transcriptional analysis of human islets obtained 3–6 days post-isolation detected insulin-positive cells with reduced expression of β-cell genes with concomitant elevated levels of progenitor markers, indicating that an early ex vivo dedifferentiation process has taken place [499]. There is, therefore, a need to develop proper culture conditions to maintain islet function both for in vitro studies and for islet preservation prior to islet transplantation. Several studies have shown that co-culture of islets with MSCs prevents the ex vivo loss of function of islets (Table 2) [463,500,501]. These observations suggest that MSCs provide factors that sustain β-cell function and survival and prevent the spontaneous dedifferentiation that usually occurs in culture. 

Yeung et al. [348] observed that human MSCs protected human islets from the destruction caused by the pro-inflammatory cytokines IFNγ, TNFα, and IL-1β. The MSC-mediated cytoprotection was attributed to the secretion of HGF and metalloproteinases 2 and 9 [348]. MMP-2 and MMP-9 have been shown to contribute to the immunosuppressive function of MSCs by reducing the surface expression of IL-2R (CD25) on T cells [502]. MMP9 knockout mice showed normal development of pancreata and islets but had an impaired response to glucose load in vivo, and MMP9 knockout islets secreted a reduced amount of insulin in response to glucose [503]. This suggests that extracellular matrix turnover is important for releasing paracrine factors from the matrix.

**Table 2 biomedicines-11-02558-t002:** In vitro evidence for β-cell supporting roles of MSCs.

In Vitro Effects of MSCs on Islet Function	References
Human MSCs cultures supported human islet function in an indirect co-culture system where the islets were separated from the MSC monolayer by a membrane. Islets exposed to MSC-secreted factors showed a decreased ADP/ATP ratio compared to islet monoculture, and their insulin-secretion function was improved. The conditioned medium of MSC contained HGF, TGFβ, IL-6, and VEGF-A. The co-culture medium had lower TNFα and IFNγ content than that of the islet monocultures.	[504]
Mouse islets cultured with human umbilical cord-derived MSCs showed increased viability, reduced ADP/ATP content, and increased glucose-stimulated insulin secretion. Islets cultured with MSCs showed increased expression of the anti-apoptotic XIAP (X-linked inhibitor of apoptosis protein).	[379]
Rat islets cultured in the absence of MSCs gradually lost their structural integrity and insulin-secretion function within the first three weeks. Rat islets incubated with rat MSCs showed preserved morphology and functional insulin secretion. The islets were surrounded by the MSCs. The MSCs prevented the production of TNFα and MCP-1 from the islet, while the TIMP-1 and VEGF levels were higher in the co-culture than in the islet monocultures.	[505]
Human islets were protected from IL-1β-induced cell death by human MSCs overexpressing hepatocyte growth factor (HGF) and interleukin-1 receptor antagonist (IL-1Ra).	[506]
Human bone marrow-derived MSCs protected human islets from apoptosis induced by the combined treatment of TNFα, IL-1β, and IFNγ.	[507]
Rat islets co-cultured 38 days together with rat adipose-derived MSCs or bone marrow (BM)-derived MSCs showed significant improvement in basal insulin secretion levels compared to islets cultivated alone. Indirect incubation of streptozotocin-damaged rat islets with rat bone-marrow-derived MSCs resulted in the survival of the islets.	[508,509]
Cultivation of islets on MSC monolayers retained their insulin-producing function and prevented the dedifferentiation of islets occurring in culture.	[63]
Rat MSCs prolonged the survival of rat islets in culture and increased glucose-stimulated insulin secretion in comparison to islets alone. MSCs in contact with pancreatic islets differentiated into cells that express PDX1 and secrete insulin. MSCs adhered to the pancreatic islets.	[463]
Treatment of streptozotocin-injured mouse islets in vitro with the conditioned medium from bone marrow-derived MSCs increased the activation of AKT and ERK1/2 in the islets. The MSC-conditioned medium increased proliferation of β-cells in injured islets in an AKT-dependent manner.	[510]
Adipose tissue-derived MSCs from mice reduced the secretion of IFNγ, IL-2, and IL-17, while increasing the secretion of TGFβ, IL-4, IL-10, and IL-13 by PHA- and islet lysate-induced T cell stimulation. The MSCs protected mouse islets from the deleterious effects of reactive splenocytes in a co-culture setting.	[511]
Human islets that have been grown together with human bone marrow-derived MSCs for 3 days showed improved glucose-stimulated insulin secretion, which was further improved by addition of theophylline. The beneficial effect of MSCs was dependent on direct contact between the islets and MSCs and required the intact microstructure of the islets. Neutralizing antibodies to N-Cadherin that is expressed in the MSCs inhibited the beneficial effects of MSCs.	[512]
Rat islets encapsulated in alginate microcapsules together with MSCs and extracellular matrix proteins of the rat pancreas showed improved insulin secretion when compared to islets alone.	[513]
Human umbilical cord-derived MSCs attenuated high glucose-induced oxidative stress of rat INS-1 β-cells. The MSCs protected rat INS-1 β-cells from high glucose-induced injury and prevented the high glucose-mediated impairment in glucose-stimulated insulin secretion. MSCs increase the expression of NRF2 (Nuclear factor erythroid 2-related factor 2) and HO-1 (heme oxygenase-1) in rat INS-1 β-cells, and the knockdown of NRF2 in INS-1 abolished the protective activity of MSCs.	[495]

### 4.2. In Vivo Evidence for β-Cell Supporting Roles of MSCs

The use of MSCs in treating T1D has been shown to be effective in regulating fibrosis and tissue regeneration [48]. Co-transplantation of MSCs with pancreatic islets is more effective than islet transplantation alone in controlling glucose serum levels in diabetic animal models (Table 3). Repeated bone marrow transplantations into mice with experimental diabetes restored normoglycemia and normalized the morphology of the pancreas [514]. Systemic administration of MSCs into diabetic mice or rats resulted in pancreatic islet regeneration, increased endogenous insulin production, reduced blood glucose levels, reduced pancreatic inflammatory processes, induction of regulatory T cells, and prevention of renal damage [494,515,516]. Although some studies showed that the paracrine function of MSCs contributes to the beneficial effects of MSCs, the efficiency of MSC conditioned medium is far less efficient than MSC cell transplantation (Table 3). The beneficial effects of MSCs were especially observed when MSCs were co-transplanted with islets into diabetic animals (Table 3).

Ianus et al. [517] observed that GFP-expressing bone marrow-derived cells that have been transplanted into lethally irradiated mice have populated Langerhans islets four to six weeks after transplantation. The GFP-positive cells isolated from the islets were found to express insulin, GLUT2, and various β-cell specific transcription factors, including PDX1 (pancreatic and duodenal homeobox 1; formerly known as Ipf1—Insulin promotor factor-1) and PAX6 (Paired box protein 6) [517]. These GFP-positive cells of the islets also responded to glucose-dependent and incretin (exendin 4)-dependent insulin secretion [517]. These findings indicate that bone marrow stem cells have the ability to differentiate into insulin-producing cells, which, in part, rely on the interaction with islets. Further studies with GFP-expressing bone marrow cell transplants into diabetic mice showed increased proliferation of insulin^-^ PDX1^+^ cells, NGN3^+^ cells, and insulin^+^ glucagon^+^ cells with stem cell characteristics in the islets [518], suggesting a pancreatic regeneration role of MSCs. The mobilization of transplanted bone marrow cells to the islets was essential for pancreatic regeneration [519].

**Table 3 biomedicines-11-02558-t003:** In vivo evidence for β-cell supporting roles of MSCs in animal models.

In Vivo Effects of MSCs on Islet Function in Animal Models	References
Bone marrow-derived stem cells reduced hyperglycemia in mice that have been made diabetic by streptozotocin. The stem cells induced proliferation of pancreatic β-cells and led to increased islet insulin content.	[520]
Transplantation of human MSCs into streptozotocin-induced diabetic mice caused an increase in pancreatic islets and β-cells producing mouse insulin with concomitant reduced blood glucose levels.	[521]
Human bone marrow stromal cells (MSCs) transfected with PDX1, NEUROD1, and NGN3 expressed insulin in vitro that was not regulated by glucose. Transplantation of these modified MSCs under the kidney capsule of streptozotocin-induced diabetic mice reduced blood sugar levels.	[522]
Bone marrow-derived MSCs transplanted into streptozotocin-induced diabetic rats reduced blood glucose levels and increased the rat body weight. Some of the transplanted MSCs have differentiated into insulin-producing cells. There were many small islets in the pancreas of the MSC-treated rats.	[523]
Transplantation of MSCs into streptozotocin-induced diabetic mice reduced the blood glucose levels, reaching nearly euglycemic values after a month, which lasted for more than 2 months. There was an increase in pancreatic islets concomitant with reduced albuminuria and normal morphology of the kidney glomeruli. This was in contrast to untreated diabetic mice that presented glomerular hyalinosis and mesangial expansion.	[515]
Treatment of diabetic NOD mice with MSCs from BALB/c mice reversed hyperglycemia in most of the mice.	[516]
Transplanting rat islets together with MSCs reduced the amount of islets required to get normoglycemia. Islets transplanted with MSCs showed a higher number of capillaries than those transplanted without MSCs.	[524]
Mouse MSCs prevent the rejection of allogeneic islet grafts by secreting matrix metalloproteinases-2 and -9 (MMP2 and MMP9), which reduce CD25 expression on responding T lymphocytes. MSCs prolonged the survival of allogeneic islet grafts.	[502]
Repeated injections of MSCs into streptozotocin-induced diabetic rats resulted in lower blood glucose levels increased insulin serum levels. Peripheral T cells of the MSC-treated diabetic rats showed a shift toward IL-10/IL-13 production and higher frequencies of CD4^+^/CD8^+^ Foxp3^+^ regulatory T cells.	[494]
Co-transplantation of syngeneic rat islets with rat bone marrow-derived MSCs into omental pouch of streptozotocin-induced diabetic rats resulted in sustained normoglycemia. Transplantation of allogenic rat islets together with MSCs together with short-term immunosuppression with cyclosporin A, increased islet graft survival and insulin expression, and induced normoglycemia. The transplantation of allogeneic islets with.MSCs led to reduced production of IFNγ and TNFα, while an increased secretion of IL-10 from T cells.	[525]
Mouse MSCs suppressed diabetogenic T cell proliferation via PD-L1 and suppressed the generation of inflammatory dendritic cells. MSC treatment of type 1 diabetes in NOD mice resulted in long-term reversal of hyperglycemia.	[526]
When Lewis rat islets were transplanted together with rat MSCs into streptozotocin-diabetic syngeneic recipients or NOD-mice, a lower number of islets were required to achieve diabetes reversal. The islets transplanted together with MSCs showed improved vascularization, and islets were surrounded by MSCs.	[527]
Transplantation of mouse islets that have been co-cultured with MSC-conditioned medium into diabetic mice led to lower blood glucose levels and increased blood vessel formation. The MSC-conditioned medium contained IL-6, IL-8, VEGF, HGF, and TGFβ.	[379]
Transplantation of mouse islets together with mouse bone marrow cells resulted in enhanced islet graft vascularization, reduced blood glucose level, and increased serum insulin level.	[528]
Transplantation of mouse islets with mouse adipose tissue-derived MSCs promoted islet graft survival and insulin function of the graft in streptozotocin-induced diabetic mice and reduced the islet mass required for reversal of hyperglycemia.	[529]
Co-transplantation of mouse islets with mouse MSCs led to better reduction of blood glucose levels in streptozotocin-induced diabetic mice in comparison to mice injected with islets alone. The islets co-transplanted with MSCs showed better vascularization. Transplantation of MSCs alone did not revert hyperglycemia in the diabetic mice.	[530]
Transplantation of human islets together with human bone marrow-derived mesenchymal stem cells (hBMSCs) overexpressing human HGF and human IL-1Ra under the kidney capsule of streptozotocin-induced diabetic non-obese diabetic/severe combined immunodeficient (NOD-SCID) reversed diabetes and reduced the number of islets required to achieving normoglycemia.	[506]
Adipose tissue-derived stem cells that have been pretreated with a mixture of hyaluronic acid, butyric acid, and retinoic acid improved the islet graft revascularization. The treated adipose tissue-derived stem cells produced higher levels of VEGF and HGF.	[531]
Human bone marrow mesenchymal stem cells (MSCs) overexpressing vascular endothelial growth factor (VEGF) reversed hyperglycemia induced by streptozotocin in NOD/SCID mice. This effect was related to a better survival of β-cells. The MSC overexpressing VEGF also differentiated into vessels and β-cell-like cells.	[532]
Transplantation of rat bone marrow-derived MSCs into the pancreas of streptozotocin-induced diabetic rats resulted in reduced blood glucose levels and increased body weight. Similar results were obtained when the MSCs were injected into the head of the pancreas, the tail of the pancreas, or the whole pancreas.	[533]
Human bone-marrow-derived MSCs expressing high aldehyde dehydrogenase (ALDH) activity improved systemic hyperglycemia in streptozotocin-treated NOD/SCID mice and augmented insulin secretion by increasing islet size and vascularization. In vitro expansion of human bone-marrow-derived MSCs prior to transplantation reduced the capacity to diminish blood glucose levels in streptozotocin-treated NOD/SCID mice at two-week intervals.	[534]
Infusion of MSCs into high-fat diet/streptozotocin-induced T2D diabetic rats ameliorated hyperglycemia, reduced insulin resistance in peripheral tissue, and promoted β-cell function when delivered at the early phase (7 days after diabetes induction) but not at the later phase (21 days after diabetes induction). The MSCs promoted the recovery of streptozotocin-induced liver and pancreas damage. MSC infusion increased GLUT4 expression and increased insulin receptor substrate 1 (IRS1) and Akt phosphorylation in skeletal muscle, adipose, and liver tissues.	[535]
Transplantation of human umbilical cord-derived MSCs that have been differentiated into insulin-producing cells into the liver of streptozotocin-induced diabetic mice resulted in reduced serum glucose levels.	[536]
Co-transplantation of human bone marrow-derived MSCs together with human islets into diabetic humanized NOD SCID gamma (NSG) mice was more efficient than islet transplantation alone in improving blood glucose and serum insulin and C-peptide levels. The MSCs increased the proportion of immunosuppressive regulator T cells and protected the islets from apoptosis induced by pro-inflammatory cytokines.	[507]
Rats that have been made diabetic by high-fat diet together with streptozotocin administration became normoglycemic after at least three times of infusion of bone marrow-derived MSCs. Serum concentrations of insulin and C-peptide were increased after the serial MSC infusions.	[537]
Transplantation of human bone marrow-derived MSCs that have been differentiated into insulin-producing cells into streptozotocin-induced diabetic nude mice maintained euglycemia for 3 months. Bone marrow-derived MSCs from both diabetic and healthy human subjects could be differentiated into insulin-producing cells.	[443]
Cytoprotection of rat pancreatic islets by human adipose-derived stem cells was increased when fibroblast growth factor-2 (FGF2) was incorporated into the fibrin gel used for the subcutaneous transplantation. Some of the stem cells had differentiated into insulin-producing cells.	[538]
Rat islets co-transplanted with adipose-derived mesenchymal stromal cells (MSCs) under the kidney capsule of streptozotocin-induced diabetic rats resulted in better recovery than islet transplants alone. The MSC co-transplanted with islets had differentiated into insulin-producing cells.	[508]
In vitro cultivation of mouse islets prior to transplantation into streptozotocin-indued diabetic mice reduces their ability to reverse diabetes. However, co-cultivation of freshly isolated islets on kidney-derived MSCs improved islet graft function.	[63]
Intravenous infusion of MSCs after intra-hepatic islet transplantation or co-transplantation of MSCs together with islets under the kidney capsule resulted in improved glucose homeostasis in a streptozotocin-induced diabetic mouse model. The co-transplantation with MSCs resulted in reduced islet apoptosis.	[539]
Injection of bone marrow-derived MSCs into streptozotocin-induced diabetic mice reduced blood glucose levels increased both the size and number of islets with a concomitant increase in β-cell mass.	[510]
Subcutaneous transplantation of rat islets that have been seeded on rat MSC sheets into streptozotocin-induced severe combined immunodeficiency (SCID) diabetic mice resulted in normoglycemia in the recipient mice.	[540]
Infusion of Wharton’s jelly-derived MSCs to NOD mice at the onset of diabetes led to normal glucose homeostasis within 6–8 days that lasted for 6 weeks. Infusion of Wharton’s jelly-derived MSCs to NOD mice prior to onset of diabetes resulted in an 8-week delay in the onset of diabetes. The MSC infusion resulted in higher fasting C-peptide levels, higher frequencies of CD4^+^CD25^+^Foxp3^+^ regulatory T lymphocytes, and lower levels of IL-2, IFNγ, and TNFα.	[541]
Intrapancreatic injection of allogeneic adipose tissue-derived MSCs into streptozotocin-induced diabetic mice decreased blood glucose levels and improved glucose tolerance. However, the MSCs only remained in the tissue for a few days.	[542]
Retro-orbital venous sinus injection of GFP-overexpressing undifferentiated Wharton’s jelly-derived MSCs into NOD mice resulted in lower blood glucose levels, improved glucose tolerance, and higher survival rates. Human C-peptide and insulin could be detected in the serum of the mice. The MSC treatment resulted in reduced levels of auto-aggressive T cells and increased levels of regulatory T cells. The splenocytes expressed lower mRNA levels of IFNγ, IL-1β, TNFα, MCP-1, while higher mRNA levels of IL-4, IL-10, IL-17, and FoxP3. Fluorescent islet-like cell clusters in the pancreas could be observed whose origin was from the human MSCs. The undifferentiated MSCs differentiated into insulin-producing cells in vivo.	[543]
Co-transplantation of allogeneic islets together with autologous MSCs into streptozotocin-induced diabetic mice delayed islet rejection and increased long-term graft function in 30% of the mice. The MSCs need to be transplanted together with the islets since systemic delivery of MSCs did not have any protective effect.	[544]
Four weeks after infusion of MSCs, these cells were found to accumulate in the pancreas of diabetic NOD mice. Infusion of MSCs reduced insulitis while increasing the amount of splenic regulatory T cells. Following MSC infusion, the plasma levels of IFNγ and TNFα were reduced, while those of TGFβ1 and IL-10 were increased. The beneficial effects of MSCs were improved by Liraglutide, a long-acting GLP-1 analog.	[545]
Streptozotocin-induced diabetic mice transplanted with a combination of syngeneic adipose tissue-derived MSCs and allogeneic islet grafts showed reduced blood glucose levels and decreased pro-inflammatory cytokine (IFNγ, IL-17A) levels, while increased regulatory cytokine (TGFβ, IL-4) levels in mononuclear blood cells.	[546]
Transplantation of betatrophin-overexpressing human adipose-derived MSCs into streptozotocin-induced diabetic mice had a better effect on blood glucose levels than regular adipose-derived MSCs. In vitro co-culture of betatrophin-overexpressing human adipose-derived MSCs with human islets induced islet cell proliferation and improved glucose-stimulated insulin secretion. The betatrophin-overexpressing human adipose-derived MSCs had stronger anti-inflammatory and anti-apoptotic effects on islets than the regular adipose-derived MSCs.	[547]
MSCs that have been exposed to hypoxia express higher levels of VEGF, IL-6, MCP1, and MMP9. These hypoxia-treated MSCs increased islet grafts in a model of streptozotocin-induced diabetes in mice with lowering of the blood glucose level.	[548]
Injection of adipose tissue-derived MSCs from healthy, T2D diabetic or *db*/*db* mice into high-fat diet and streptozotocin-induced T2D diabetic mice improved insulin sensitivity and reduced β-cell death.	[549]
Transplantation of rat islets together with rat MSCs into streptozotocin-induced diabetic rats increased serum insulin levels and reduced the number of islets required for reducing blood glucose level and survival. The TNFα serum level was significantly reduced in the diabetic rats following MSC—islet co-transplantation.	[550]
Adipose tissue-derived MSCs reduced the number of islets required to achieve normoglycemia. The MSCs increased islet revascularization and the expression of angiogenic factors such as HGF and angiopoietin-1.	[551]
Administration of human telomerase (hTERT)-overexpressing MSCs into 50% pancreatectomized NMRI nude mice resulted in increased proliferation of pancreatic β-cells. The MSCs increased EGF, GLUT2, INS1, and INS2 mRNA levels in the pancreas while reducing the mRNA levels of IFNγ and TNFα.	[552]
Repeated administration of the conditioned medium of 2D and 3D human umbilical cord-derived MSCs cultures into streptozotocin-induced diabetic rats increased the rate of β-cells in the islets and increased the serum insulin and C-peptide levels. However, the conditioned medium was not sufficient to reduce the blood glucose levels in the diabetic rats. The conditioned medium increased the percentage of splenic regulatory T cells.	[553]
Transplantation of insulin-producing cells (IPCs) differentiated from human HLA-A2-negative adipose tissue-derived MSCs into the kidney capsule of streptozotocin-induced diabetic humanized mice normalized the blood glucose levels with detectable circulating human, but not mouse, insulin.	[451]
Human umbilical cord-derived MSCs in diabetic mice supported β-cell function with better insulin secretion performance. There was an increase in both the size and number of islets, and the ratio of insulin-producing β-cells was increased.	[495]
The combined treatment of streptozotocin-induced diabetic rats with bone-marrow-derived MSCs and hesperetin (a citrus flavonoid) improved glucose, insulin, and C-peptide levels.	[554]

Most of the human studies involved the transplantation of MSCs alone or in combination with mononuclear cells (MNCs) to T1D or T2D patients, which showed promising beneficial effects in terms of improved glucose homeostasis and reduced insulin requirements (Table 4). Autologous MSC transplantation in recent onset T1D patients has been shown to improve glycated HbA1c and C-peptide levels, preserve β-cell function, and shift serum cytokine patterns from pro-inflammatory cytokines to anti-inflammatory cytokines [482,488]. These beneficial effects of MSCs on islets are combined effects of direct supportive effect of MSCs on islet function and regeneration, anti-inflammatory activities, vascularization, protection of islets from hypoxic damage, and differentiation of MSCs into insulin-producing cells [517,538,555,556,557]. So far, MSC transplantation has mainly been performed in diabetic patients without co-transplantation with islets (Table 4). It is expected that islet co-transplantation with MSCs should improve the outcome in humans, as has been shown in animal studies (Table 3). 

## 5. The Paracrine Function of MSCs

Several proteomic studies have been performed to clarify the composition of the secretome of MSCs [580,581,582]. The complex secretome of MSCs consists, among others, of cytokines, chemokines, growth factors, extracellular matrix components, and extracellular vesicles [379,413,434,439,463,491,555,580,583,584,585,586,587,588,589,590,591,592,593,594] (Table 5, Figure 1 and Figure 2, and Appendix A). Some of these factors are expressed in sub-population of MSCs, and their expression levels can be affected by interaction with other cell types, by cytokines, and by hypoxia [424,595,596,597,598,599,600]. Moreover, the secretome of rat adipose-derived MSCs differs from that of rat bone-marrow-derived MSCs [601], and there are differences in the MSC secretome between different species. Despite these differences, outstanding is the secretion of VEGF, angiopoietin-1, angiogenin, activin A, FGF7, HGF, TGFβ1, stromal cell-derived factor 1 (SDF1), platelet-derived growth factor (PDGF), MCP1, TSP1, TSG14, TIMP1, IL-8, IL-6, CXCL1, and IGFBP3 [379,494,509,511,524,581,584,585,602]. Rat adipose tissue-derived MSCs express higher levels of IL-1α, IL-6, CXCL1, CCL20, and CCL2 than rat bone-marrow-derived MSCs, while the latter express higher levels of Wnt1 inducible signaling pathway protein 2 (WISP2), osteomodulin (OMD), TGFβ2, and BMP4 [601].

Some of the MSC-produced factors support β-cell function and survival, as mentioned in Table 1. Trophic factors produced by MSCs with a beneficial effect on β-cell survival and function include VEGF [377,524,585], CNTF [270,603], HGF [348], von Willebrand factor [524,525,527,528,603], SDF-1 [220], and IL-6 [379]. SDF1 (CXCL12) has a positive effect on β-cell differentiation and survival besides causing immunosuppression and promoting wound repair [220,371,372,604]. MSCs promote angiogenesis by virtue of the secretion of bFGF and VEGF as well as certain cytokines such as IL-1, IL-6, and M-CSF (CSF1) [605]. The pro-angiogenic VEGF, which is highly expressed both in MSCs [379] and islets [263], has been shown to act as a survival factor for human islets [377]. Human islets that have been cultured in MSC-conditioned medium expressed higher levels of anti-apoptotic signal molecules (X-linked inhibitor of apoptosis protein (XIAP), Bcl-xL, Bcl-2, and heat shock protein-32 (HSP32)) and increased expression of vascular endothelial growth factor receptor 2 (VEGFR2) [379]. Altogether, the production of several different growth factors by MSCs may explain how co-administration of islets with MSCs can improve the efficiency of islet transplantation [63,606]. 

An important property of MSCs is their ability to survive under hypoxic conditions [598,607,608,609]. Exposure of MSCs to hypoxic conditions enhances the expression of VEGFA, PDGF, bFGF, IL-10, IL-6, IL-8, RANTES, MCP-1, TGFβ and MMP9 [548,599,600,608,610]. Analogously, stimulation of MSCs with TNFα increases their secretion of the pro-angiogenic cytokines IL-6 and IL-8 and the chemokines CXCL5, CXCL6, CXCL10, and MCP1 [582,611] as well as the growth factors VEGF, HGF and insulin-like growth factor I (IGF-I) [612]. IL-6, IL-8, and MCP1 are also involved in monocyte chemoattraction [582]. The production of HGF by adipose tissue-derived MSCs is stimulated by bFGF and EGF [613]. TNFα-stimulated MSCs promote endothelial progenitor cell homing and angiogenesis [611]. The production of heme oxygenase (HO)-1 by MSCs protects islets from injury caused by hypoxia and reoxygenation [614]. Pro-inflammatory cytokines reduce HO-1 expression in rat islets, which is prevented by the co-culture with human MSCs [615]. Thus, the cytoprotective and angiogenetic effects of MSCs are enhanced under hypoxia and inflammatory conditions.

The production of TGFβ1, indoleamine 2,3-dioxygenase (IDO), prostaglandin E_2_ (PGE_2_), IL-10, HGF, metalloproteinases, HO-1, tumor necrosis factor-induced protein 6 (TSG6), and nitric oxide (NOˑ) by MSCs has been associated with their immunosuppressive properties [62,348,502,616,617,618,619,620,621,622]. Some of these factors are induced in MSCs by inflammatory cytokines. For instance, IFNγ induces MSC expression of IDO, which catalyzes the conversion of tryptophan to kynurenine, resulting in tryptophan depletion and suppression of T lymphocyte function by metabolites of kynurenine [618,623]. Nitric oxide (NO·) production by MSCs, which is also induced by IFNγ, suppresses STAT5 phosphorylation and T cell proliferation [620]. The combined treatment of MSCs with both IFNγ and IL-1β induces a higher expression of IDO and PGE_2_ than each cytokine alone, resulting in better immunosuppressive activities as demonstrated in a murine colitis model [624]. IFNγ also induces Programmed death-ligand 1 (PD-L1, B7-H1, CD274) expression on MSCs that further suppresses T cell proliferation [625]. The secretion of the chemokines CCL2 and CXCL12 by MSCs contributes to the polarization of macrophages into IL-10-producing cells involved in anti-inflammatory responses [626]. PD-L1 and PD-L2 expression is upregulated in MSCs under inflammatory conditions, which contribute to immunosuppression by interacting with PD-1 receptors on T lymphocytes [442,627,628]. Stimulation of MSCs with IFNγ and TNFα induces the secretion of PD-L1 that suppresses the activation of CD4^+^ T cells and downregulates IL-2 secretion [628]. Other studies show that MSCs increase a subpopulation of CD4^+^ that produces IFNγ and IL-10, an effect that depends on IFNγ-stimulation of the MSCs [629].

Extracellular vesicles produced by MSCs have also been shown to contribute to their immunosuppressive activities [583,630,631]. Among others, these vesicles prevent antigen uptake by immature dendritic cells and the maturation of dendritic cells with reduced expression of the activation markers CD83, CD38, and CD80 and decreased secretion of the pro-inflammatory cytokines IL-6 and IL-12p70 while increased production of the anti-inflammatory cytokine TGFβ [632]. By using dendritic cells differentiated from CD14^+^ cells isolated from T1D patients, MSC-derived extracellular vesicles were found to induce regulatory dendritic cells, resulting in reduced IFNγ secretion by interacting T cells and the appearance of FOXP3^+^ regulatory T cells [425]. Favaro et al. [633] further showed that MSC-derived extracellular vesicles were internalized by peripheral blood mononuclear cells isolated from T1D patients and prevented T lymphocyte activation following stimulation with the islet antigen glutamic acid decarboxylase. The vesicles also resulted in a shift in the cytokine profile with increased levels of TGFβ, IL-10, IL-6, and PGE_2_ [633]. Exosomes from adipose tissue-derived MSCs ameliorated autoimmune reaction in a streptozotocin-induced T1D mouse model with elevated levels of IL-4, IL-10, and TGFβ and concomitantly reduced levels of IL-17 and IFNγ [634]. Treating obese mice fed on a high-fat diet with MSC-derived extracellular vesicles resulted in increased glucose uptake and alleviation of insulin resistance [635].

MSC-derived extracellular vesicles have also been shown to have potential therapeutic applications in regenerative medicine [630,631,636,637,638,639,640] and, as such, they have been incorporated in several clinical trials [630]. The vesicles carry with them many of the bioactive molecules produced by MSCs and have the advantage of being cell-free, non-replicating, and showing low immunogenicity [630,631]. The small size of the vesicles allows them to be taken up by recipient cells through pinocytosis, resulting in alterations in their functionality and activities [631]. Among human diseases that have been treated with MSC-derived extracellular vesicles include acute respiratory distress syndrome (ARDS), wounds, and inflammatory diseases such as Crohn’s disease, ulcerative colitis, and periodontitis [630]. MSC-derived extracellular vesicles have further been shown to ameliorate diabetic foot ischemia and ulcer [641], diabetic nephropathy [642,643,644], and other diabetic-related complications [645].

**Table 5 biomedicines-11-02558-t005:** The secretome of MSCs.

Secreted Factor	Effects Associated with the MSC Secreted Factors *	References
Differentiation factors e.g., Activin A, BMP4, BMP6, TSP1	Activin A expression in MSCs is required for their chondrogenic and osteogenic differentiation. Activin A production by MSCs induces neuronal development and neurite outgrowth. Activin A and betacellulin, which are produced by MSCs, are involved in β-cell differentiation during development and β-cell regeneration in adults. Treating human islets with activin A led to reduced expression of genes associated with β-cell maturity (e.g., PDX1, MAFA, GLUT2), while increased genes expressed in immature β-cells (e.g., MAFB). Bone morphogenic protein 4 and 6 (BMP4 and BMP6) secreted by MSCs can promote differentiation of adipose tissue-derived MSCs. BMP4 has also been used to differentiate MSCs into keratinocytes. BMP2 and BMP4 stimulate the chemotactic migration of human mesenchymal progenitor cells. The roles of BMP2, BMP4, and BMP6 in β-cell differentiation and function are described in Table 1. Thrombospondin-1 (TSP1) is responsible for the activation of the latent secreted form of TGFβ1. Treating TSP1 knockout mice with a peptide derived from TSP1 led to TGFβ1 activation and reversal of the pancreatic abnormalities observed in the TSP1 knockout mice.	[230,258,375,581,584,594,646,647,648,649,650,651,652,653]
Chemokines, e.g., CXCL1, CCL2 (MCP1), CCL5 (RANTES), CCL7, CXCL4, CXCL5, CXCL12 (SDF-1), CXCL16; CCL22, eotaxin 2 (CCL24) and eotaxin 3 (CCL26), CCL28, Fractalkine (CX3CL1)	MSCs express a whole series of chemokines, among them CCL2, CXCL1, and CXCL5 are outstanding. CCL2, which recruits macrophages, is expressed at low levels in resting MSCs, but it is highly upregulated by inflammatory cytokines such as TNFα. CCL2 secreted from MSCs promotes the polarization of macrophages to an M2 neuroprotective phenotype. CCL2 and CXCL12 (SDF1) form heterodimers that induce IL-10 expression in CCR2-expressing macrophages, thus mitigating gut injury caused by dextran sulfate sodium (DSS)-induced colitis in mice. Stroma-derived factor-1 (SDF1/CXCL12) secreted from MSCs contributes to tissue regeneration and repair. SDF1 (CXCL12) regulates mobilization of neutrophils from the bone marrow. The MSC-derived SDF-1 increases the phagocytic function of neutrophils, resulting in better clearance of bacteria. Chemokines are also involved in the migration of MSCs to wounds and injured tissues.	[584,586,590,594,604,626,654,655,656,657]
Cytokines, e.g., IL-1α, IL-1β, IL-4, IL-6, IL-8, IL-10, GM-CSF, G-CSF, M-CSF	Human bone marrow-, human umbilical cord- and cord blood-derived MSCs express several cytokines, with the most prominent ones being IL-6 and IL-8. IL-4 is secreted by human islets, human Wharton’s Jelly MSCs, and mouse bone marrow-derived MSCs. IL-4 receptor is expressed on human islet cells. IL-4 protects β-cells from IFNγ/IL-1β-induced apoptosis by activating the PI3K and JAK/STAT pathways. IL-6 protects MIN6 β-cells against the pro-apoptotic signals delivered by IL-1β, IFNγ, and TNFα. Lipopolysaccharide (LPS) stimulates the secretion of GM-CSF, G-CSF, and M-CSF from adipose tissue-derived MSCs, which contributes to hematopoiesis. TNFα stimulation of human adipose tissue-derived MSCs resulted in the upregulation of IL-6, IL-8, and MCP-1, which stimulate the migration of monocytes.	[203,205,379,544,581,582,584,590,613,658]
Growth and survival factors, e.g., EGF, FGF6, FGF7, bFGF (FGF2), HGF, IGF2, PDGF-AA, PDGF-AB, PDGF-BB, VEGF, BDNF, GDF15, TSP1, adiponectin, TGFβ, SCF	Human and mouse MSCs secrete a whole battery of growth factors. GDF15 secreted from human umbilical cord blood-derived MSCs promotes neurogenesis and increases amyloid β-clearance by microglial cells. Thrombospondin-1 (TSP1) secreted by MSCs attenuates amyloid β peptide-induced synaptic dysfunction. PDGF-BB stimulates the chemotactic migration of human mesenchymal progenitor cells. Deletion of insulin-like growth factor 2 (IGF2) in pancreatic mesenchymal-derived cells results in acinar and β-cell hypoplasia. The effects of the various growth factors on β-cell function are described in Table 1.	[222,544,581,584,590,594,595,604,653,659,660,661]
IGFBPs	Insulin-like growth factor binding proteins (IGFBP) 1,2, 3, 4, and 6 are expressed in MSCs. IGFBPs are involved in regulating the effects of IGFs on growth, development, and metabolism by binding to IGF1 and IGF2. The anti-diabetic effect of leptin is in part mediated through induction of IGFBP2. Overexpression of IGFBP2 reversed diabetes in insulin-resistant *ob*/*ob* mice and streptozotocin-induced diabetic mice. IGFBP2 overexpression improved hepatic insulin sensitivity in *ob*/*ob* mice. IGFBP1 increased the number of cells in mouse and human islets that co-express glucagon and insulin. IGFBP1 promoted β-cell regeneration in Zebrafish. High IGFBP1 levels in humans reduced their risk of developing T2D.	[581,584,594,662,663]
Neurotrophic factors, e.g., BDNF, CNTF, βNGF, GDNF, NT4, NRG1	Human MSCs express several neurotrophic factors, including BDNF, CNTF, βNGF, Glial derived neurotrophic factor (GDNF), and neurotrophin 4 (NT4). Rat adipose tissue-derived MSCs express higher levels of neuroregulin 1 (NRG1) than rat bone marrow-derived MSCs. MSCs promote neurite outgrowth within dorsal root ganglion explants, which seems to be a combined effect of several secreted neuro-regulatory molecules since the βNGF levels in the MSC supernatant is far below the concentration required for inducing neurite outgrowths. Islets which are highly innervated express receptors for nerve growth factors. The effects of the neurotrophic factors on β-cell function are described in Table 1.	[270,584,594,595,601,603]
Factors involved in tissue regeneration, e.g., bFGF, EGF, GM-CSF, IGF, TSG6 and TSG14	MSCs promote wound closure in mouse wound models. MSCs promote cutaneous wound repair in diabetic mice. Through secretion of trophic factors such as bFGF, EGF, GM-CSF, and IGF, MSCs can promote the survival of stem and progenitor cells in its vicinity, resulting in improved tissue repair. TSG6 and TSG14 (Pentraxin-3/PTX-3) are involved in tissue repair and wound healing. TSG6 mediates the anti-fibrotic effects of MSCs on macrophages. TSG6 suppresses TNFα secretion from activated macrophages and induces a switch from a high fibrotic to a low anti-fibrotic TGFβ1/TGFβ3 ratio. TGFβ1 secreted by macrophages upon phagocytosis of neutrophils, terminates inflammation and induces myofibroblast differentiation. Together with enhanced release of TIMP1, TGFβ1 leads to wound and tissue fibrosis. TGFβ3, on the other hand, has anti-scarring activity, reduces early extracellular matrix deposition, and increases IL-10 production by macrophages. MSCs can further promote wound healing in mice by undergoing trans-differentiation into different skin cell types, including keratinocytes.	[581,584,602,652,664,665,666,667,668,669,670]
Pro-angiogenetic factors, e.g., VEGFA, VEGFB, VEGFC, VEGFD, Angiopoietin-1, Angiopoietin-2, Angiogenin, IGF-1, Netrin-1, HGF, IL-6, IL-8, MCP-1, CXCL16, PDGF, MMP8 and MMP9	MSCs promote vascular remodeling and angiogenesis by secreting a variety of pro-survival and angiogenic factors such as VEGF, Angiopoietin-1, Angiogenin, IGF-1, Netrin-1, HGF, IL-6, IL-8, MCP-1, CXCL16, PDGF, MMP8 and MMP9. TNFα activates MSCs to secrete the pro-angiogenic cytokines IL-6 and IL-8. The conditioned medium from TNFα-activated MSCs stimulated blood perfusion and angiogenesis when injected into the ischemic hindlimb of a mouse model. MMP9 knockout mice showed similar islet mass and distribution as wild-type mice but had an impaired glucose response in vivo with lower serum insulin levels. MMP9 knockout islets also showed reduced glucose-stimulated insulin secretion in vitro. The vascular density of the MMP9 knockout islets is reduced, with the capillaries having fewer fenestrations. Angiopoietin-1 (ANG1) and angiopoietin-2 (ANG2) stimulate islet-like development from human induced pluripotent stem cells (iPSCs). Netrin-1 produced by human Wharton’s jelly MSCs, belongs to a family of laminin-like proteins that interact with DCC/Neogenin-1 and UNC5 receptors to stimulate or inhibit angiogenesis, depending on the context. Islet grafts co-transfected with MSCs showed increased vascularization and were surrounded by von Willebrand factor-expressing endothelial cells.	[238,379,503,524,527,528,529,581,584,585,586,591,594,608,611,671,672,673,674]
Immunosuppressive factors, e.g., HGF, PGE_2_, IDO, TGFβ1, TGFβ3, GILZ, Activin A, IL-6, IL-10, nitric oxide, HO-1, TSG6, TSG14, VEGF, STC-1, PD-L1, MMP2 and MMP9	The secretion of HGF by MSCs suppresses T-lymphocyte proliferation. The secretion of TGFβ1 by MSCs suppresses T-lymphocyte proliferation. Secretion of indoleamine-2,3-dioxygenase (IDO) by MSCs, which is upregulated by IFNγ, inhibits the proliferation of activated T and NK cells by converting tryptophane into kynurenine. Human MSCs express higher levels of IDO than mouse MSCs, while mouse MSCs express higher levels of iNOS than human MSCs. The immunosuppressive function of MSCs is induced by IFNγ. TNFα and IFNγ act synergistically to induce PGE_2_ production in MSCs, which in turn suppresses immune cells. PGE_2_ secreted by MSCs polarizes macrophages to an M2 phenotype, producing the immunosuppressive cytokine IL-10. PGE_2_ secreted from MSCs reduces TNFα secretion while increasing IL-10 secretion from dendritic cells. Human umbilical cord-derived MSCs produce activin A and PGE_2_, which co-operate in suppressing IFNγ production by NK cells. IL-6 produced by MSCs increases the secretion of PGE_2_ from MSCs. Wild-type, but not IL-6-deficient MSCs, could alleviate the clinical signs of collagen-induced arthritis in mice. The MSCs modulate the host response by inducing a switch from a Th1/Th17 towards a Th2 immune profile. MSC-derived IL-6 inhibits the differentiation of dendritic cells as well as preventing the proliferation of T cells. MSC-derived IL-10 inhibits Th17 cell differentiation. Stimulation of MSCs with IL-10 induces the secretion of sHLA-G, which has immunosuppressive activities. Nitric oxide production by MSCs suppresses T cell proliferation. The expression of chemokines leads to the attraction of T cells to the MSCs where they become suppressed by nitric oxide produced by MSCs. Heme oxygenase -1 (HO-1) produced by MSCs co-operates with iNOS to induce immunosuppression. GILZ (glucocorticoid-induced leucine zipper or TSC22D3) produced by bone-marrow MSCs promotes regulatory T cells. When MSCs are primed with IFNγ and TNFα, GILZ translocates to the nucleus, where it stimulates the transcription of iNOS and activin A. Activin A, in turn, represses Th17 cells and enhances IL-10 production. TGFβ has immunosuppressive activities and is a major contributor to the immunosuppressive function of MSCs. TGFβ acts together with HGF to mediate the immunosuppressive function of MSCs. The exposure of MSCs to IL-4 and/or IL-13 increases their production of TGFβ. Activin A promotes the TGFβ-mediated conversion of CD4^+^CD25^-^ T cells into Foxp3^+^ regulatory T cells. VEGF has anti-inflammatory activities on activated T cells while increasing the number of regulatory T cells. 3D MSC cultures secrete tumor necrosis factor-stimulated gene 6 (TSG6), and stanniocalcin 1 (STC1). Tumor necrosis factor-α-induced gene/protein 6 (TSG6) promotes wound healing and contributes to an immunosuppressed environment, among others, by inducing M1 to M2 macrophage polarization. TSG6 interacts with CD44 receptor on macrophages to reduce zymosan/TLR2-mediated nuclear translocation of NFκB. TSG6 is also involved in reducing inflammation by MSCs under condition of myocardial infarction in an animal model. TSG6 mediates the anti-inflammatory effects of human MSCs in a mouse model of LPS-induced lung inflammation. TSG6 mediates the anti-inflammatory activity of human MSCs in a mouse model of severe acute pancreatitis. TSG14 (also called penetraxin 3, PTX3) binds to apoptotic cells and prevents their recognition by dendritic cells. NLRP3-activated macrophages stimulate human MSCs to secrete stanniocalcin 1 (STC1), which, in turn, inhibits NLRP3 inflammasome activation and ROS production in macrophages. Knockdown of stanniocalcin 1 (STC1) in MSCs prevents the suppression of chimeric antigen receptor (CAR)-T cells caused by MSCs. In addition, stanniocalcin 1 (STC-1) has anti-apoptotic activity. Secretion of PD-L1 and PD-L2 from MSCs, which is induced by IFNγ and TNFα, causes T cell immunosuppression. MMP2 and MMP9 secreted by MSCs cause immunosuppression by reducing CD25 surface expression on responding T cells. Inhibition of MMP2 and MMP9 prevented the MSC-mediated attenuation of delayed-type hypersensitivity responses to allogeneic antigens and restored T cell responses to IL-2 in a mouse model. Inhibition of MMP2 and MMP9 prevented the MSC-mediated protection of allogeneic islet grafts in a model of streptozotocin-induced diabetic mice.	[409,410,427,442,502,581,584,594,596,597,618,619,620,621,622,627,628,666,675,676,677,678,679,680,681,682,683,684,685,686,687,688,689,690,691,692,693,694,695,696,697,698,699,700,701,702,703,704,705,706,707]
Antioxidant factors, e.g., HO-1	Heme oxygenase-1 (HO-1) produced by MSCs exhibits antioxidant properties and protects islets from hyperglycemia-induced oxidative stress. HO-1 acts via the Nuclear factor erythroid 2-related factor 2 (NRF2) signaling pathway.	[495]
Other factors secreted by MSCs	GIF or migration inhibitory factor (MIF) protects from ischemic injury and cellular apoptosis. Leptin is strongly induced in MSCs upon hypoxia. Leptin controls appetite and energy balance and regulates secondary metabolism, cell survival, migration, and angiogenesis. Endostatin is an internal fragment of collagen XVIII that inhibits angiogenesis. Serpin E and Serpin F are serine proteinase inhibitors that control the activities of proteases involved in inflammation, complement, coagulation, and fibrinolytic pathways. Amphiregulin binds to EGF receptor and stimulates cell growth, survival, and migration. Amphiregulin promotes tissue repair. Amphiregulin reduces ER stress in cultured islets. Amphiregulin is upregulated in islet-infiltrated regulatory T cells. Osteostatin M inhibits osteoclast activity and differentiation. RANKL (receptor activator of NFκB ligand) negatively regulates osteoblastic bone formation. TIMP1, TIMP2, and TIMP4 inhibit MMPs and thus regulate tissue repair.	[524,525,527,581,584,590,594,708,709,710]

* β-cell related functions are described in Table 1.

Dietrich et al. [590] studied the cytokine content in supernatants of mono- or co-culture of human islets and Wharton’s jelly MSCs. This study showed a higher expression of IL-1β, IL-17, IFNγ, IL-4, IL-10, IL-13, Granulocyte-macrophage colony-stimulating factor (GM-CSF), and leptin in the supernatant of the co-cultures than in islet or MSC monocultures [590]. They also observed that human islets secrete various chemokines, including CCL2, CCL3, CCL4, and GROα (CXCL1), and to a lesser extent, CCL5. Wharton’s jelly MSC monocultures secreted much higher levels of CCL4 than islets, while lower levels of CCL2 and GROα (CXCL1) than the islets [590]. Both the human islets and human MSCs produce adiponectin, which was not further upregulated in the co-cultures [590]. 

To better understand the protective activity of MSCs on islet function, it was intuitive to look for similarities and differences in the growth factor profile of human MSCs and human islets using a human growth factor RT Profiler PCR Array. This study showed that there are growth factor genes that are expressed in both MSCs and islets, while others are more prominent in one cell type in comparison to the other (Figure 1 and Figure 2). Among the genes expressed in both human MSCs and human islets, we could find *BMP1*, *CSF1*, *FGF2*, *FGF14*, *IGF2*, *INHBA*, *MDK*, *PDGFC*, *PGF*, *SPP1*, *TGFB1*, *VEGFA*, and *VEGFC* (Figure 1 and Figure 2A). Genes that are highly expressed in islets, with relatively low levels in MSCs include *BMP5*, *BMP8b*, *CECR1*, *CXCL1*, *FGF13*, and *LEFTY1* (Figure 1 and Figure 2B). The human islets also express the cytokines IL-11, IL-18, IL-1α, and IL-1β (Figure 1 and Figure 2B). On the other hand, genes predominantly expressed in human MSCs with relatively low expression of human islets include *BDNF*, *DKK*, *FGF5*, *FGF7*, *IGF1*, JAG1, *NGF*, NRG1, *PTN*, *LTBP4*, and *NDP* (Figure 1 and Figure 2C).

BMP1 is a procollagen C-proteinase that has been shown to promote osteogenesis of bone marrow-derived MSCs [711]. BMP1-like proteases are also involved in the activation of growth factors by cleaving BMP2, BMP4, GDF11, and TGFβ1 [712]. In addition, it cleaves both human and mouse IGFBP3, thereby reducing the ability of IGFBP3 to bind and block IGF1 [713]. The bone morphogenic protein BMP5 has previously been shown to be exclusively expressed in β-cells among islet cells [263,264,265]. BMP5 has been implicated in the development of fetal pancreatic epithelium [266].

The growth factor array showed that MSCs express several FGF genes including FGF2, FGF5, FGF7, and FGF14. Among them, FGF2 and FGF7 have been used in β-cell differentiation protocols (e.g., [235]). FGF2 (basic FGF) is a notochord factor that represses endodermal sonic hedgehog, thereby permitting the expression of the pancreatic genes PDX1 and INS (insulin) [300]. FGF7, also known as keratinocyte growth factor (KGF), has been shown to lead to ductal cell differentiation into β-cells [299], and it is included in the differentiation protocol of human pluripotent stem cells into β-cells [235]. FGF10 has been previously described as a mesenchymal factor that promotes the development of pancreatic epithelium [301]. FGF14 has also previously been shown to be produced by MSCs [714] and mouse islets [715] and might play a role in fine-tuning neuronal function [716]. 

VEGF expression is important for the highly developed vascularization in islets, which is crucial for the rapid endocrine responses to variances in glucose blood levels [378]. VEGF also acts as a survival factor for human islets [377]. VEGF has repeatedly been shown by other research groups to be expressed in both MSCs [581,585,594,599,605,612] and islets [72,717,718,719]. CSF1 (M-CSF), which supports the differentiation and survival of monocytes and macrophages, can induce the polarization of macrophages to a pro-angiogenic M2 phenotype [720,721].

Two INHBA (inhibin βA) subunits form the homodimeric activin A, which is a differentiation factor affecting β-cell differentiation, β-cell regeneration, and glucose-stimulated insulin secretion [227,228,229,230,722] (Table 1). In some studies, activin A and TGFβ1 have been included in the early steps of β-cell differentiation in vitro. However, at later differentiation stages, inhibition of the TGFβ/activin/nodal and BMP pathways was required for induction of *PDX1* and *INS* gene expression [226]. Follistatin, which is also expressed in islets [723], inhibits the activin A-mediated down-regulation of *PDX1*, *MAFA*, and *GLUT2* in a mouse β-cell line [648].

MSCs were found to produce several neurotrophic factors, including Midkine, BDNF, NGF, NRG1, and PTN (pleiotrophin). Taking into account the similarities between neuron and β-cell evolution despite being derived from different germ layers [724], it is likely that the MSC-secreted neurotrophic factors might have a beneficial role in insulin-producing β-cells. Midkine (neurite growth-promoting factor 2, MDK) is a heparin-binding cytokine that promotes the growth, survival, and migration of target cells such as neural precursor cells [725]. Pleiotrophin (PTN) is another heparin-binding cytokine with neurotrophic activities [725]. In mice at the mRNA level, PTN is especially expressed in immature β-cells with low GLUT2 expression [726]. The PTN peptide was detected in adult mouse islets with a predominant presence in β-cells [726]. In the embryonic pancreas, PTN appears in areas of blood vessel formation near differentiating ductal epithelium [727]. Inhibition of PTN expression in mouse embryonic pancreatic primordia explants prevented full maturation of endocrine precursors with impaired insulin and glucagon expression [727]. Further studies suggest a role for PTN in β-cell proliferation and regulation of glucose homeostasis [728,729]. BDNF has been shown to bind to the TrkB.T1 receptor on β-cells, resulting in increased GSIS [247]. There are also some lines of evidence that NGF may fine-tune insulin secretion through acting on the TrkA receptor expressed in β-cells [224,225]. β-cells might themselves transiently secrete NGF upon glucose stimulation [224,225].

Many differentiation protocols of embryonic stem cells (ESCs) and induced pluripotent stem cells (iPSCs) have included various growth factors such as GDF8, FGF7, FGF10, activin A, FGF2, BMP4, HGF, IGF1, Wnt3a, and FGF7 (KGF), in attempts to obtain insulin-producing β-cells [233,234,235,730,731,732,733,734,735,736,737,738,739,740,741]. The choice of growth factors has, in general, been based on the knowledge of growth factors required for normal pancreatic development [730,742,743]. Most of these studies have the limitation of low percentage of mature β-cells and short-term maintenance. Our observation that MSCs can sustain islet function in vitro suggests that MSCs ought to be included in the β-cell differentiation protocols.

## 6. Conclusions

In this review, we have addressed various aspects of immune-mediated β-cell destruction and cytokine-induced β-cell death leading to T1D diabetes, as well as the β-cell protective roles of MSCs and their potential clinical applications in regulating glucose homeostasis with reduced insulin requirement and even insulin independence in selected early onset diabetic people. Another important topic discussed is the MSC secretome of diverse cytokines, chemokines, growth factors, angiopoietic factors, and immunosuppressive factors, which collectively influence various aspects of diabetes pathogenesis, ultimately providing a microenvironment that promotes β-cell differentiation, growth, and survival, and protects the β-cells from the hazardous effects of immune cells and pro-inflammatory cytokines. The protective effects of MSCs on β-cells are a combination of β-cell differentiation, maintenance of mature β-cell functions, β-cell growth and survival, regulation of insulin secretion including glucose-induced insulin secretion, angiogenesis, protection against hypoxia-induced and cytokine-induced β-cell damage, and immunosuppression (Figure 3). An additional component is the ability of MSCs by themselves to differentiate into insulin-producing cells when encountering islets and their attraction to injured and inflamed areas. The human studies have so far focused on the transplantation of bone marrow monocytes and MSCs with or without concomitant immunosuppressive therapy, usually resulting in a transient beneficial effect with some diabetic people reaching long-term effects. Considering the data obtained from animal studies, it would be desirable to combine the MSCs with islet transplantation. However, the sparse amount of human islets available for this purpose is a limitation. The increased knowledge of the MSC secretome can be used in further studies to optimize the growth factor composition for preserving β-cell function and to increase the efficiency of in vitro β-cell differentiation. Another recommendation would be to use MSCs to preserve the ex vivo function of islets.

## Figures and Tables

**Figure 1 biomedicines-11-02558-f001:**
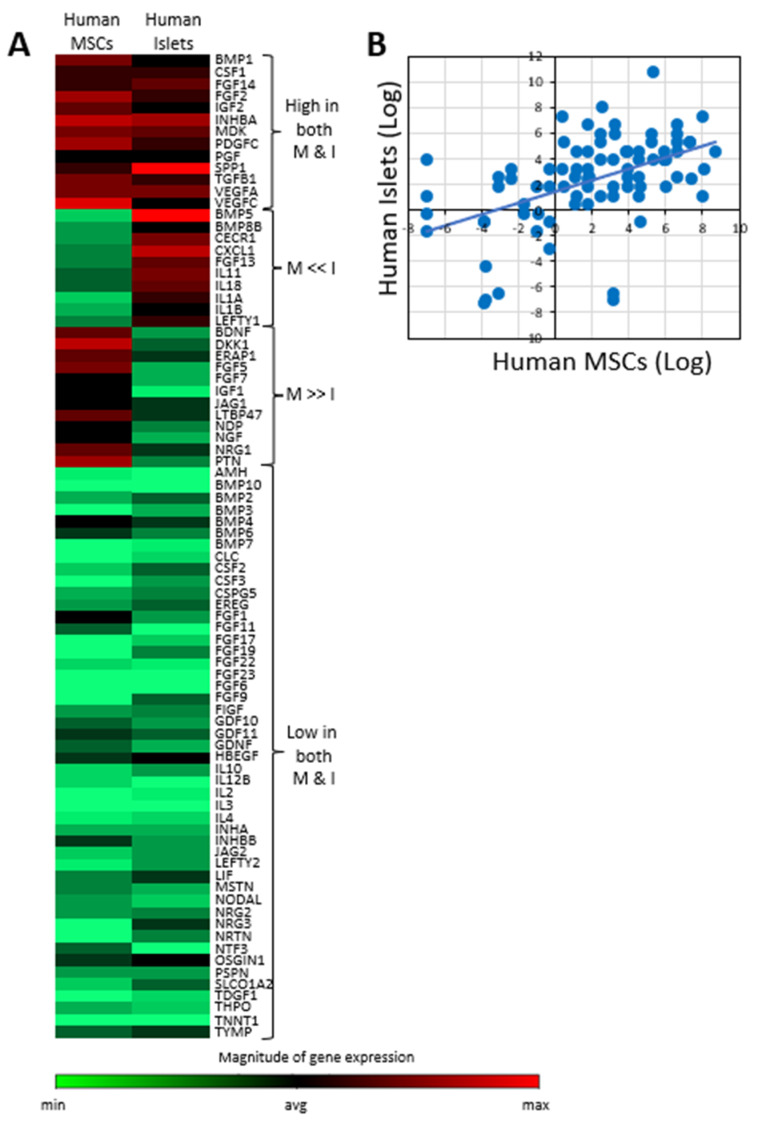
(**A**). Heatmap visualization of growth factor and cytokine mRNA expression in human bone marrow-derived MSCs (Lonza, Catalog number PT-2501, Walkersville, MD, USA) and human islets (obtained from PRODO Laboratories Inc., Irvine, CA, USA) using the Human growth factor RT Profiler PCR Array PAHS-041A (Qiagen, MD, USA). M = Mesenchymal stem cells. I = Islets. (**B**). A clusterogram of growth factor genes expressed in human islets versus human MSCs.

**Figure 2 biomedicines-11-02558-f002:**
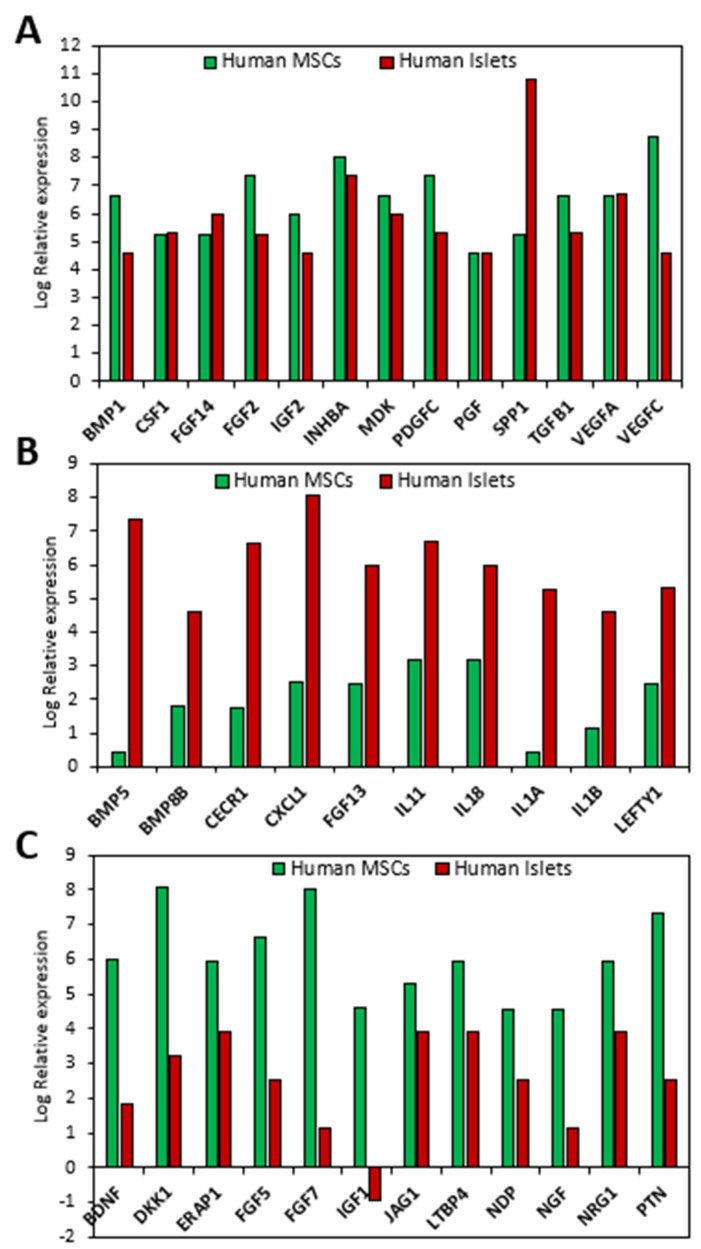
(**A**–**C**). mRNA expression in human bone marrow-derived MSCs and human islets for the indicated genes as determined using the Human growth factor RT Profiler PCR Array as shown in Figure 1.

**Figure 3 biomedicines-11-02558-f003:**
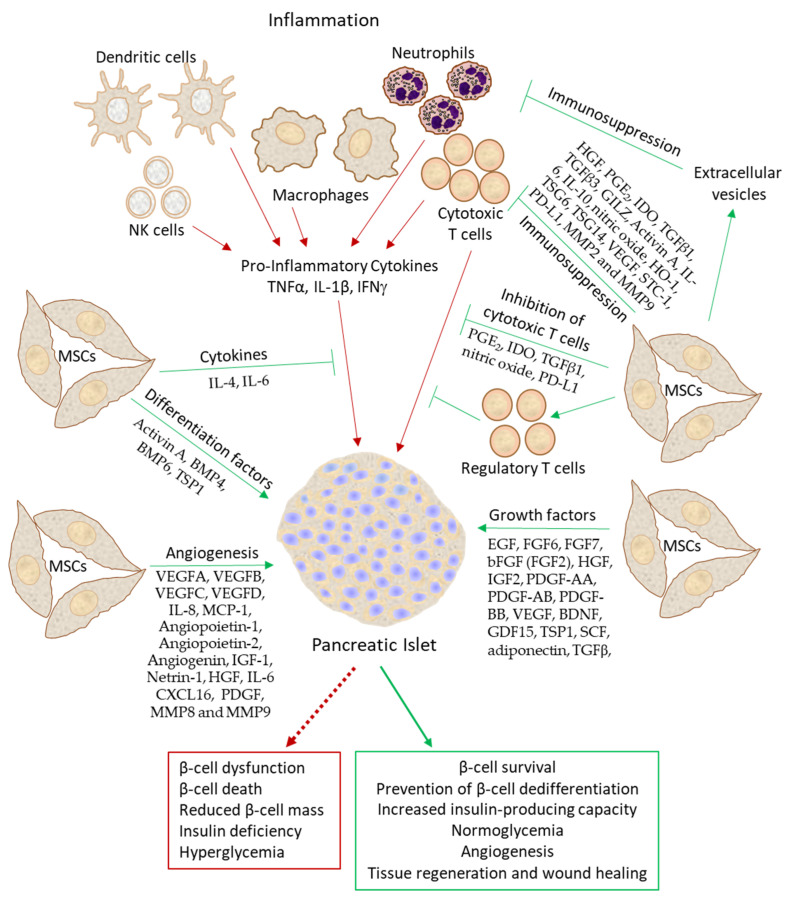
An illustration of the beneficial effects of MSCs on pancreatic β-cells. The red arrows show the deleterious effects of inflammation on β-cells, whereas the green arrows show the beneficial effects of MSCs on β-cells, including the prevention of the harmful effects of cytotoxic T cells and inflammatory cytokines.

**Table 1 biomedicines-11-02558-t001:** Growth factors supporting β-cell survival and function.

Growth Factor	Effect on β-Cell Function	References
Activin A	Activin A could be detected by immunostaining at an early embryonic stage of rat pancreatic development. Activin A is expressed in fetal rat pancreas and human β-cells. Activin A, which belongs to the transforming growth factor-β (TGF-β) family, stimulates insulin secretion in cultured human and rat pancreatic islets when incubated in the presence of glucose. Activin A phosphorylates SMAD2 and SMAD3. Activin A, in combination with betacellulin, lowered the serum glucose concentration of newborn Sprague-Dawley rats that have been made diabetic with streptozotocin. The number of islets was increased by the combined treatment of activin A and betacellulin. Activin A and TGFβ1 have been included in the early steps of β-cell differentiation of stem cells in vitro. However, at later differentiation stages, inhibition of the TGFβ/activin/nodal and BMP pathways was required for induction of *PDX1* and *INS* gene expression. The activin B receptor ALK7 was found to be a negative regulator of pancreatic β-cells. When TGFβ signaling was disrupted in adult mouse β-cells by conditional overexpression of SMAD7 in PDX1^+^ cells, the mice developed diabetes, which could be reversed upon resumption of islet TGFβ signaling. However, when SMAD7 was activated in PDX1^+^ mouse β-cells during embryonic development, β-cell hypoplasia and neonatal lethality occurred.	[226,227,228,229,230,231,232,233,234,235,236,237]
ANG1 and ANG2	Angiopoietin 1 (ANG1) and angiopoietin 2 (ANG2) promote the generation of all hormone-producing cells of the islets (including insulin, glucagon, somatostatin, and pancreatic polypeptide) from human induced pluripotent stem cells (iPSCs). Both systemic ANG1 knockout mice and β-cell-specific ANG1 knockout mice showed reduced serum insulin levels and glucose intolerance, suggesting an essential role of angiopoietin-1 for normal islet function.	[76,238]
BDNF	Brain-derived neurotrophic factor (BDNF) reduces glucagon secretion while increasing glucose-induced insulin secretion from mouse islets. BDNF receptor TrkB.T1 is expressed on β-cells. TrkB.T1 knockout mice showed impaired glucose tolerance and insulin secretion. BDNF is secreted from differentiated human muscle cells and thus might be a mediator to increase glucose metabolism during exercise. Repeated administration of BDNF to obese diabetic *db*/*db* mice reduces blood glucose concentration. Repeated BDNF treatment of *db*/*db* mice increases the islet insulin content. BDNF increases the hepatic glucokinase levels and improves the response to insulin in obese insulin-resistant rats. BDNF suppresses hepatic glucose production. BDNF reduces food intake.	[239,240,241,242,243,244,245,246,247]
BMP2, BMP4, BMP5 and BMP6	Both Bone morphogenetic protein 2 (BMP2) and BMP4 are expressed in islets. BMP4 is expressed in islet ε-cells, with very low levels in α, β, γ, or δ-cells. During mouse pancreatic development, BMP4, BMP6, BMP7, and TGFβ1 were detected in E13.5, E15.5, and E17.5 fetal mouse pancreas, while BMP2, BMP5 and activin A were detected only at the later E17.5 stage. BMP4, BMP5, and BMP6 promoted the formation of cystic colonies from dissociated E15.5 mouse pancreatic cells, which required the presence of laminin 1. The activity of these BMPs was antagonized by activin A and TGFβ1. BMP2 is upregulated by pro-inflammatory cytokines in rodent and human islets. BMP2 and BMP4 are upregulated in animal models of T1D and T2D. BMP5 is exclusively expressed in β-cells. Both BMP2 and BMP4 play important roles during β-cell development. The BMP signaling acts within a timely temporal window during embryonic pancreatic development to either promote or prevent β-cell differentiation. In a Zebrafish model, BMP signaling is required for the formation of ventral pancreatic cells, but when exposed to BMP signaling, they do not develop into β-cells. Inhibition of BMP signaling at the late embryonic stage by using the BMP receptor inhibitor dorsomorphin resulted in increased β-cell neogenesis near the extrapancreatic duct. Thus, BMP signaling is required in specific time windows during β-cell differentiation. BMP4 induces the expression of Inhibitor of DNA binding (Id) proteins that bind to the basic helix-loop-helix (bHLH) transcription factor NeuroD, required for embryonic differentiation of islets. Thus, BMP4 blocks the differentiation of endocrine progenitor cells while promoting their expansion. The WNT ligand WNT5A improves β-cell differentiation, among others, by cooperating with Gremlin 1 to inhibit the BMP pathway during β-cell maturation. BMP2 and BMP4 inhibit basal and growth factor-stimulated proliferation of primary β-cells from adult rats and mice. Glucose-induced insulin secretion was impaired in adult rodents and human islets pre-treated with BMP4. The islet pericytes are important for β-cell function, among others, through production of BMP4. Transcription factor 12 (TCF12) of the canonical Wnt signaling is required for BMP4 secretion from pancreatic pericytes. Inactivation of pericytic TCF12 in mice results in impaired glucose tolerance, comprising β-cell function and glucose-stimulated insulin secretion. Exogenously added BMP4 rescues the impaired glucose-induced insulin secretion. In a transgenic mouse model, BMP4 promotes the expression of core β cell genes and enhances glucose-stimulated insulin secretion and glucose clearance. The BMP4 receptor BMPR1A (BMP type 1a receptor or ALK3) is expressed on β-cells, and mice with attenuated BMPR1A signaling in β-cells develop diabetes due to impaired insulin secretion. The β-cells from these mice express lower levels of genes involved in insulin gene expression and show impaired glucose-stimulated insulin secretion. High glucose increases the expression of BMP2 and BMP4 in human aortic endothelial cells, resulting in vascular inflammation. Noggin, a bone morphogenic protein inhibitor, which is often co-upregulated with BMPs, reduces serum glucose levels in *db*/*db* T2D mice. BMP4 is one of the growth factors used to differentiate human embryonic stem cells and human induced pluripotent stem cells into definitive endoderm cells. Treatment of leptin-deficient *ob*/*ob* mice (T2D model) with BMP6 led to reduced blood glucose and lipid levels while increasing plasma insulin levels.	[134,146,248,249,250,251,252,253,254,255,256,257,258,259,260,261,262,263,264,265,266]
CCK	Cholecystokinin (CCK) is a gut hormone that is also expressed in islets. CCK is upregulated in obesity. CCK production and secretion is induced in β-cells by GLP-1. Loss of CCK results in increased β-cell apoptosis in obese mice. CCK promotes β-cell survival in rodent models of diabetes. CCK reduces cytokine-mediated apoptosis of β-cells and human and mouse islets.	[267,268,269]
CNTF	Ciliary neurotrophic factor (CNTF) reduces glucose-induced insulin secretion of rat islets. CNTF promotes islet cell survival by upregulating Connexin 36, PAX4, and Bcl-2. CNTF, together with EGF, protects mice from alloxan-induced hyperglycemia by increasing the number of β-cells.	[270,271]
CTGF	Connective tissue growth factor (CTGF) is highly expressed in islet vasculature and interacts with TGFβ and Wnt signaling pathways. CTGF contributes to β-cell proliferation and development in mice. CTGF is expressed in endothelial cells, pancreatic ducts, and embryonic β cells. Inactivation of endothelial CTGF leads to decreased islet vascularity and decreased embryonic β-cell proliferation. Overexpression of CTGF in β-cells promotes proliferation of immature embryonic β-cells.	[272,273,274,275]
EGFs	Exposure of human islets to epidermal growth factor (EGF) leads to dedifferentiation into duct-like epithelial structures. Dedifferentiation is important for the EGF-induced proliferation, for later to undergo redifferentiation. EGF receptor (EGFR) signaling is essential for proper fetal development of islets and for sufficient β-cell mass post-partum. EGFR signaling is required for β-cell mass expansion during high-fat diet and during pregnancy in mice. Combined treatment of human islets with EGF and gastrin in vitro increased the number of β-cells as well as the insulin content of the cultured islets. EGF as well as TGFα transiently induced phosphorylation of ERK1/2, GSK3 and AKT (PKB) in the INS-1 rat β-cell line. EGF and gastrin restored normoglycemia and induced islet regeneration in alloxan-induced diabetic mice. Treatment of alloxan-induced diabetic mice with EGF and ciliary neurotrophic factor (CNTF) led to an increased number of insulin-positive cells and normalization of blood glucose levels. Treatment of rat exocrine pancreatic cells with EGF and leukemia inhibitory factor (LIF) resulted in transdifferentiation into insulin-producing cells expressing C-peptide, PDX1, and GLUT2. Transgenic mice overexpressing EGF and keratinocyte growth factor (KGF) under the insulin promoter showed profound alterations in pancreatic morphology with an enlargement of the islets, profound intra-islet fibrosis, and appearance of intra-islet duct cells. Heparin-binding epidermal growth factor (EGF)-like growth factor (HB-EGF) stimulates β-cell proliferation of rat and human islets. Heparin-binding epidermal growth factor (EGF)-like growth factor (HB-EGF) mRNA levels are increased in β-cells in response to glucose. Epiregulin, a member of the EGF-related growth factor family, stimulates proliferation of rat β-cell lines through activation of EGFR/ErbB1. Epiregulin increases glucose uptake in *Lep^ob^* mice by binding to leptin receptors, resulting in translocation of the glucose transporter GLUT4 to the cell surface. Betacellulin, a member of the epidermal growth factor family, is produced by proliferating pancreatic β-cells and induces the differentiation of a pancreatic acinar cell line into insulin-secreting cells, an effect enhanced by activin A. In a dorsal embryonic pancreas culture from day 11.5 embryos, betacellulin enhances branching morphogenesis and increases PDX1 and insulin production while it inhibits the production of amylase and glucagon. Betacellulin increases the number of β-cells in islets of streptozotocin-treated mice. Betacellulin ameliorates hyperglycemia in obese leptin receptor-deficient *db*/*db* mice (T2D model).	[226,271,276,277,278,279,280,281,282,283,284,285,286,287,288,289,290,291,292,293,294,295,296]
FGFs	Fibroblast growth factors (FGFs) are required for β-cell differentiation during pancreatic development. FGF receptor (FGFR) 1 and 2, and the ligands FGF1, FGF2 (basic FGF, bFGF), FGF4, FGF5, FGF7 (keratinocyte growth factor, KGF), and FGF10 are expressed in adult mouse β-cells. FGF7/KGF leads to ductal cell differentiation into β-cells. FGF7/KGF promotes β-cell regeneration by stimulating duct cell proliferation. Overexpression of FGF7/KGF in acinar tissue leads to islet hyperplasia. Mice overexpressing FGF7/KGF under the proximal elastase promoter showed an increased number of islets and increased islet size at the average. Overexpression of FGF7/KGF in β-cells under the insulin promoter caused the islets to become fibrotic. FGF7/KGF treatment of mice that have been transplanted with human fetal pancreatic cells led to an increased number of β-cells in the graft. FGF7/KGF serum level increases with age. FGF10 promotes the development of pancreatic epithelium. FGF21 increased insulin levels in normal rat islets but did not potentiate glucose-induced insulin secretion. However, in islets from diabetic rats, FGF21 increased both islet insulin content and glucose-induced insulin secretion. FGFR1c signaling is required for the expression of pro-insulin convertases. Mice with defective FGFR1c signaling developed diabetes with age and showed a lower β-cell mass. FGF2 and FGF7/KGF have been used in β-cell differentiation protocols. FGF2 (basic FGF) is a notochord factor that represses endodermal sonic hedgehog (SHH), thereby permitting the expression of the pancreatic genes PDX1 and INS (insulin).	[164,235,297,298,299,300,301,302,303]
GDF11	Growth differentiation factor 11 (GDF11) is expressed in embryonic islet progenitor cells that express neurogenin 3 (NGN3), and promotes β-cell differentiation during pancreas development. GDF11-deficient mice have an increased number of NGN^+^ islet progenitor cells, reduced β-cell numbers, and impaired β-cell maturation.	[304,305]
GDF15(MIC-1)	Growth/differentiation factor 15 (GDF15/MIC-1) expression is suppressed under inflammatory conditions and in patients with T1D diabetes. Obese people have increased plasma GDF15 concentrations, with the highest concentrations observed in T2D patients. T2D individuals show increased GDF15 (MIC-1) serum levels. IL-1β and IFNγ suppress GDF15 mRNA in the islets. Exogenously added GDF15 protects the islets from IL-1β and IFNγ-induced apoptosis. GDF15 prevents insulitis and decreases incidence of diabetes in NOD mice. GDF15 improves insulin sensitivity in mice fed with a high-fat diet. GDF15 decreases body weight and increases insulin sensitivity in *ob*/*ob* mice, which is attributed to elevated oxidative metabolism and lipid mobilization in liver, muscle, and adipose tissue. GDF15 is an anti-inflammatory cytokine that decreases TNFα production in lipopolysaccharide-activated macrophages. Especially high expression of GDF15 is found in the acinar and ductal cells of the exocrine pancreas. GDF15 is upregulated by the ER stress CHOP transcription factor and the Th2 cytokines IL-4 and IL-13. GDF15 is required for the IL-13-induced improvement of glucose intolerance in mice fed with a high-fat diet.	[39,252,306,307,308,309,310,311,312,313]
GH	Growth hormone (GH) stimulates proliferation and insulin production of β-cells through a mechanism involving the activation of Janus kinase 2 (JAK2)/ Signal transducer and activator of transcription 5 (STAT5) signaling pathway. GH activation of STAT5 protects β-cells from cytokine (IFNγ, TNFα, IL-1β)-induced apoptosis. Growth hormone and IGF-1 synergistically increase the survival and proliferation of β-cells.	[210,314,315,316,317]
GIP	Gastric inhibitory polypeptide (GIP; also termed glucose-dependent insulinotropic polypeptide) is an incretin that is released by enteroendocrine K-cells after meal ingestion and promotes survival and proliferation of β-cells, as well as potentiating insulin secretion. Activation of the GIP receptor leads to stimulation of adenylyl cyclase and Ca^2+^-independent phospholipase A2 and activation of PKA and PKB. GIP increases the expression of the anti-apoptotic Bcl-2 and decreases the expression of the pro-apoptotic Bax. T2D patients do not respond to the insulinotropic activity of GIP.	[318,319,320,321,322,323,324,325,326]
GLP-1	Glucagon-like polypeptide-1 (GLP-1) is an insulinotropic intestinal-derived incretin that is released after meal ingestion and promotes survival and proliferation of rodent β-cells. GLP-1 enhances insulin secretion and reduces glucagon secretion. GLP-1 increases cholecystokinin (CCK) production in β-cells, which protects them from cytokine-induced apoptosis. Exendin-4 is a GLP-1 analog resistant to cleavage by dipeptidyl peptidase 4 (DPP-IV) and acts as a GLP-1 receptor agonist to increase cAMP intracellular levels and improve glycemic control in T2D patients. GLP-1 and exendin-4 stimulate β-cell neogenesis in streptozotocin-treated newborn rats. Exendin-4 promotes the differentiation and maturation of human fetal pancreatic cells. Exendin-4 protects against cytokine-induced β-cell death by increasing connexin 36 gap junction levels on the plasma membrane.	[269,318,319,320,321,325,327,328,329,330,331,332,333,334,335]
HGF	Hepatocyte growth factor (HGF) promotes fetal β-cell proliferation and proliferation of mature β-cells in pregnancy. Fetal pancreas-derived fibroblasts express HGF, which stimulates β-cell proliferation and islet cluster formation. HGF, together with activin A, induces the differentiation of a pancreatic acinar cell line into insulin-secreting cells. HGF increases islet engraftment and islet transplant performance in diabetic rodents. The beneficial effects of HGF seem to be both due to the protection of β-cells from cell death and promotion of their proliferation. The HGF-induced proliferation of β-cells was increased by low glucose concentrations (3–6 mM). HGF activates the JAK2/STAT5 and phosphatidylinositol-3’-kinase (PI3K)/AKT pathways. Cultivation of human islets in fibrin gel together with HGF preserved β-cell function and increased engraftment in a mouse model. Overexpression of HGF in the β-cell of adult transgenic mice results in increased β-cell mass, β-cell proliferation, and β-cell survival. These mice were more resistant to the diabetogenic effects of streptozotocin. Injection of an HGF-expressing plasmid prior to streptozotocin treatment of mice attenuated diabetes development with a slower increase in blood glucose levels and maintenance of higher serum insulin levels. These effects seem to be due to protection of the islets from the β-cytotoxic effects of streptozotocin. The HGF treatment increased pro-survival Akt kinase activation and Bcl-xL expression in the islets of the mice. Disruption of the HGF/c-Met signaling in mice results in increased β-cell death and early onset of diabetes in a mice model of multiple low-dose streptozotocin administration.	[336,337,338,339,340,341,342,343,344,345,346,347,348,349]
IGF1 and IGF2	Insulin-like growth factor (IGF1) is important for maintaining normal glucose homeostasis. IGF1 promotes β-cell survival and proliferation and decreases β-cell apoptosis. Treatment of rat islets with IGF1 prevented IL-1β-induced nitric oxide production and the consequent reduction in islet cell death. IGF1 activates ERK1/2, GSK3 and the PI3K/AKT signaling pathway in INS-1 rat β-cells. Mice with specific deficiency of the IGF1 receptor in β-cells showed normal β-cell mass but had reduced expression of GLUT2 and glucokinase, resulting in defective glucose-stimulated insulin secretion and impaired glucose tolerance. IGF2 is a survival signal for β-cells. IGF2 is expressed in fetal and neonatal rat islet cells but declines rapidly 2 weeks after birth, which is associated with increased neonatal islet cell death. Transgenic mice overexpressing IGF2 led to increased islet mass and prevented the apoptosis of islet cells seen in normal mice between postnatal days 11 and 16. IGF2 prevents cytokine (IFNγ, IL-1β, TNFα)-induced islet cell death.	[215,216,217,296,350,351,352]
INGAP	Islet neogenesis-associated protein (INGAP) protects β-cells from cytokine-induced apoptosis. INGAP, which is expressed in pancreatic acinar cells, can stimulate islet production from pancreatic progenitor cells. Transgenic mice overexpressing INGAP in pancreatic acinar cells are resistant to hyperglycemia induced by streptozotocin.	[212,353,354,355]
NGF	Nerve growth factor (NGF) is expressed in the pancreatic vasculature, and its TrkA receptor is found on β-cells. High glucose concentration increases NGF secretion and activates the TrkA receptor, resulting in increased insulin secretion both in mouse and human islets. NGF augments glucose-induced insulin secretion. Tissue-specific deletion of NGF or TrkA receptor in mice impaired glucose tolerance and insulin secretion. It is proposed that NRG fine-tunes insulin secretion by maintaining low basal insulin secretion while increasing glucose-stimulated insulin secretion.	[224,225]
NRGs	Neuroregulin (NRG)1α is expressed in β-cells, while NRG1β and NRG3 are mainly found in α-cells. NRG3 increased the proliferation of a rat insulinoma cell line. NRG4 increased the insulin secretion from a rat insulinoma cell line. NRG4 is an adipokine that increases glucose metabolism in peripheral organs.	[356,357]
OPN	Osteopontin (OPN) protects islets and β-cells from IL-1β-mediated cytotoxicity and streptozotozin-induced β-cell death by negatively regulating nitric oxide production. Osteopontin inhibits the synthesis of inducible nitric oxide synthase (iNOS). IL-1β and IFNγ treatment of human islets led to downregulation of osteopontin (SPP1), while streptozotocin treatment of mice led to an upregulation of osteopontin. The increased osteopontin expression may be a feedback mechanism to counterbalance the effects of pro-inflammatory cytokines. Osteopontin improved glucose-stimulated insulin secretion in mildly diabetic rat islets and in human islets from cadaver diabetic donors but not of islets from cadaver normoglycemic donors. Osteopontin serum level increases with age. High glucose and incretins stimulate islet osteopontin secretion.	[39,164,219,326,358,359]
PDGF-AA	Platelet-derived growth factor AA (PDGF-AA) isoform promotes β-cell proliferation in human juvenile β-cells that express PDGF receptor, while it does not affect adult human β-cells that do not express the PDGF receptor. PDGF-AA promotes proliferation of β-cells and improves their insulin-secretion function. The serum and tissue level of PDGF-AA, which is produced by osteoblasts, decreases with age.	[360,361]
PIGF	Placental growth factor (PIGF) is expressed in β-cells of adult mouse pancreas. PIGF levels are increased in β-cells during pregnancy. Knocking down PIGF in β-cells resulted in reduced β-cell proliferation and impaired glucose tolerance in pregnant mice.	[362]
PL-I	Placental lactogen I (PL-I) promotes β-cell proliferation, increases the β-cell mass, and results in hypoglycemia in a transgenic model overexpression of this gene.	[363]
Prolactin	Prolactin stimulates proliferation and insulin production of β-cells. Prolactin activates the JAK2/STAT5 pathway and phosphatidylinositol-3’-kinase (PI3K). Prolactin upregulates the expression of Survivin, which promotes β-cell proliferation.	[314,315,364,365]
PTHrP	Parathyroid hormone-related protein (PTHrP) induces insulin expression by activating MAP kinase-specific phosphatase-1 that dephosphorylates JNK. PTHrP is present in the pancreatic islet. An N-terminal PTHrP peptide (1–36) stimulates β-cell proliferation and preserves β-cell function in adult mice.	[366,367,368,369,370]
SDF-1/CXCL12	Stromal cell-derived factor 1 (SDF-1; also known as C-X-C motif chemokine 12 (CXCL12)) promotes β-cell development and survival. It is expressed in developing β-cells and during adult β-cell regeneration but repressed in terminally differentiated mature β-cells. SDF-1 enhances glucose-stimulated insulin secretion by human pluripotent stem cells and induces β-cell specific genes in these stem cells. SDF-1 protects islets from cytokine-induced apoptosis. SDF-1 causes immunosuppression.	[220,371,372]
TSP1	Thrombospondin 1 (TSP1) protects β-cells from lipotoxicity by activating the PKR-like ER kinase (PERK)—nuclear factor erythroid-2-related factor-2 (NRF2) signaling pathway involved in producing a protective antioxidant defense response. The anti-apoptotic growth factor thrombospondin is downregulated by IL-1β and IFNγ in human islets. TSP1-deficient mice are glucose intolerant despite having an increased β-cell mass. Their islets showed decreased glucose-stimulated insulin release, insulin biosynthesis, and glucose oxidation rate. One of the positive effects of TSP1 on pancreatic islet morphology is mediated by the activation of TGFβ1. TSP1 is mainly expressed in the endothelium of the normal islet. TSP1-deficient mice are glucose intolerant despite showing increased β-cell mass.	[39,373,374,375]
VEGF	Microvasculature plays an important role for intact islet function. Human islets produce several VEGF isoforms and express VEGF receptors 1, 2, and 3 as well as the co-receptor Neuropilin 1. Vascular endothelial growth factor (VEGF) affects both vascularization and innervation of the pancreatic islets. VEGF production by pancreatic islets is essential for islet vascularization and function. Mice with reduced VEGF-A expression in β-cells showed impaired glucose-stimulated insulin secretion. The highly developed vascularization in islets is essential for the endocrine responses to variances in glucose homeostasis. VEGF also acts as a survival factor for human islets. VEGF prevents human islet death induced by serum starvation. Transplantation of VEGF-overexpressing mouse islets into streptozotocin-induced syngeneic diabetic mice improved glycemic control already at day 1 post-transplantation, suggesting increased survival of the islet graft. The immunosuppressive drug rapamycin reduces islet VEGF secretion as well as islet viability and insulin release. Transplantation of rat islets together with vascular endothelial cells carrying a VEGF-expressing plasmid into diabetic rats restored blood glucose and insulin levels more efficiently than islets alone.	[72,376,377,378,379,380]

**Table 4 biomedicines-11-02558-t004:** Evidence for beneficial effects of hematopoietic stem cells and MSCs in diabetic patients.

Effects of Stem Cell Treatment in Diabetic Patients	References
In newly diagnosed T1D patients who received high-dose immunosuppression followed by autologous nonmyeloablative hematopoietic stem cell transplantation (AHST), 14 out of 15 patients became insulin-free for a period of 6–35 months. Patients with diabetic ketoacidosis failed to benefit from autologous nonmyeloablative hematopoietic stem cell transplantation. Adverse effects were observed in one patient who developed acute culture-negative bilateral pneumonia, and 2 patients appeared with late endocrine dysfunction (hypothyroidism or hypogonadism).	[558]
Intraportal administration of human adipose-tissue-derived, insulin-producing MSCs together with unfractionated cultured bone marrow to five insulinopenic T1D patients resulted in a 30–50% reduction in insulin requirement with a 4–26-fold increase in serum C-peptide levels, as analyzed after 1.5 to 3.5 months after treatment.	[485]
Autologous nonmyeloablative hematopoietic stem cell transplantation (HSCT; prepared by cyclophosphamide and granulocyte colony-stimulating factor (G-CSF)-mediated mobilization into the blood) to 23 newly diagnosed T1D patients without prior ketoacidosis resulted in insulin-free condition in 20 of the individuals. 12 patients remained insulin-free for 14–52 months, and of these, 8 patients relapsed and required low insulin doses. Patients who benefited from HSCT experienced an increase in C-peptide levels. Two patients developed bilateral nosocomial pneumonia, 3 patients developed late endocrine dysfunction, and 9 patients developed oligospermia.	[559]
A case report of a man with early-onset T1D who received 4 doses of cyclophosphamide along with anti-thymocyte globulin followed by autologous hematopoietic stem cell transplantation. This treatment led to normoglycemia that lasted more than 5 months after transplantation.	[560]
Co-transplantation of human adipose tissue-derived insulin-secreting mesenchymal stem cells and cultured human bone marrow into 11 male T1D patients resulted in reduced exogenous insulin requirement, increased serum C-peptide levels, and no diabetic ketoacidosis. The differentiation of adipose tissue-derived MSCs into insulin-producing cells was done by exposure the cells to nicotinamide, activin A, exendin 4, pentagastrin, HGF, B-27, and N-2 supplements.	[484]
Eight patients with early diagnosed T1D underwent immunoablation (high-dose cyclophosphamide and anti-thymocyte globulin) and hematopoietic stem cell transplantation (mobilized with cyclophosphamide and G-CSF). All patients became insulin-free after transplantation. One of them required a low insulin dose after 7 months, and six of the patients received the anti-diabetic drug acarbose (an alpha-glucosidase inhibitor) for better glycemic control.	[561]
Human placenta-derived MSCs were infused three times into 10 T2D patients at one-month intervals. The daily insulin requirement was reduced, and the serum C-peptide levels were increased along with reduced glycosylated hemoglobin. Also, the renal and cardiac functions were improved after MSC infusion.	[562]
Autologous hematopoietic stem cell transplantation (AHSCT) into 13 newly diagnosed T1D patients, among them 10 with diabetic ketoacidosis, resulted in lower insulin requirement in 11 of the individuals, accompanied by decreased IL-1, IL-17, and TNFα serum levels, increased C-peptide concentration and reduced glycosylated hemoglobin. Three patients become insulin-free for 7–54 months.	[563]
28 T1D patients underwent autologous nonmyeloablative hematopoietic stem cell transplantation after pretreatment consisting of a combination of cyclophosphamide and anti-thymocyte globulin. Insulin independence was achieved in 15 out of 28 T1D patients over a period of 4 to 42 months, with a much higher response rate in patients without diabetic ketoacidosis (70.6% responding) than those suffering from diabetic ketoacidosis (27.3% responding).	[564]
A case report of autologous hematopoietic stem cell transplantation (obtained by cyclophosphamide and G-CSF mobilization to blood) that was delivered to T1D patients with diabetic ketoacidosis together with subcutaneous administration of G-CSF to increase the peripheral blood neutrophil count. The insulin requirement was gradually reduced, and reached insulin independence after 27 days, which lasted more than 70 months. The treatment with cyclophosphamide was accompanied by nausea, vomiting, fever, alopecia, and leukopenia.	[565]
15 Newly onset T1D patients who were treated with Wharton’s jelly-derived MSCs showed a gradual reduction in blood glucose levels, increased fasting C-peptide levels, and reduced insulin requirement that lasted for more than 24 months. 4 of the 15 patients became insulin independent. 2 patients were non-responders. The HbA1c level was gradually reduced. The transplantation with Wharton’s jelly-derived MSCs does not require previous immunosuppression with cyclophosphamide and anti-thymocyte globulin, which is of great advantage.	[489]
Intrahepatic autologous bone marrow stem cell transplantation (stimulated with filgrastim (G-CSF)) was performed in two recently diagnosed T1D patients with ketoacidosis. These two patients showed increased levels of C-peptide and reduced blood glucose and HbA1c levels for at least 12 months of follow-up. The serum levels of anti-islet antibodies were strongly reduced by the stem cell transplantation.	[557]
Repeated transfusion of umbilical cord-derived MSCs resulted in increased C-peptide levels and increased number of regulatory T cells in a subgroup of T2D patients.	[566]
Autologous bone marrow-derived stem cell transplantation was introduced twice to 11 T2D patients. Nine out of the 11 (82%) reached a reduction of more than 50% in insulin requirement up to 12 months, with increased stimulated C-peptide serum levels.	[567]
Transplantation of hematopoietic stem cells to 65 newly onset T1D accompanied with immunosuppression therapy (anti-thymocyte globulin and cyclophosphamide) resulted in insulin independence in 59% of the patients for the first 6 months, and 32% remained insulin-independent after 48 months. All treated individuals showed a decrease in serum HbA1c levels and an increase in serum C-peptide levels.	[568]
Infusion of in vitro-generated donor bone marrow-derived hematopoietic stem cells to a 5-year-old T1D patient improved glucose homeostasis for 6 months. Infusion of in vitro-generated donor bone marrow-derived hematopoietic stem cells, together with autologous adipose tissue-derived MSCs that have been differentiated into insulin-producing cells, to a 9-year-old T1D patient resulted in stable blood glucose levels with reduced insulin requirement.	[569]
Two treatment groups were enrolled in this study. Group 1 included 10 T1D patients who received autologous MSC therapy from their own adipose tissue and bone marrow, while Group 2 included 10 T1D patients who received allogeneic MSC therapy obtained from healthy, compatible, non-diabetic volunteer donors. The adipose-derived MSCs were differentiated into insulin-producing cells using the protocol involving nicotinamide, activin A, exendin, pentagastrin, HGF, B27, and N2. Both groups showed reduced insulin requirement, reduced serum HbA1c levels, increased C-peptide serum levels, and reduced fasting blood glucose levels. Some of these parameters were significantly better in the autologous than in the allogeneic group. The MSC treatment also prevented episodes of diabetic ketoacidosis. These data show that adipose tissue-derived MSCs from T1D patients can be differentiated into insulin-producing cells that can be used to treat diabetes of the same individual.	[570]
Early onset T1D patients receiving autologous MSC transplantation showed preserved or increased C-peptide blood levels in response to a mixed-meal tolerance test (MMTT), indicative of preserved β-cell function.	[488]
T1D patients transplanted with umbilical cord-derived MSCs together with autologous bone marrow mononuclear cell stem cells showed increased serum insulin levels and reduced HbA1c levels. The fasting glucose level was reduced concomitant with a reduced insulin requirement.	[487]
24 T1D patients underwent immunoablation and autologous hematopoietic stem cell transplantation (AHSCT). 20 of the patients remained insulin-free for at least 9.5 months, with four remaining insulin-free for 34–80 months. No severe complications were observed, except for one patient who died of pseudomonas sepsis in the course of neutropenia after transplantation.	[571]
16 T1D patients got autologous hematopoietic stem cell transplantation following immunosuppressive regimens. 7 patients achieved insulin independence, six showed reduced insulin requirement, while 3 did not respond.	[572]
4 T1D patients with ketoacidosis were treated with bone marrow-derived MSCs. Two of the patients showed reduced insulin requirement, and one became insulin-independent for 3 months. The fourth patient remained stable on the same insulin dose for 4 years.	[573]
The immune responses after autologous hematopoietic stem cell transplantation were studied in 18 T1D patients. Patients who have received autologous hematopoietic stem cell transplantation showed increased fasting C-peptide levels, reduced serum HbA1c, and reduced insulin requirement. The Th1 cells from the peripheral blood of the stem cell transplanted patients secreted lower levels of IL-2, IL-12p40, and IFNγ. Stem cell transplantation reduced the proportion of Th17 cells while increasing the amount of regulatory T cells.	[556]
6 out of 10 T2D patients transplanted with either bone marrow-derived MSCs or mononuclear cells (MNCs) showed reduced insulin requirement. MSCs increased peripheral insulin sensitivity.	[574]
20 T1D patients received autologous hematopoietic stem cell transplantation. 14 of the patients developed insulin independence for 1.5 to 48 months. Thereafter, they returned to regular insulin use.	[575]
Transplantation of autologous MSCs to 5 T1D patients resulted in lower insulin requirements and increased leptin serum levels.	[576]
27 T1D patients received infusion of allogeneic umbilical cord-derived MSCs with a repeated MSC infusion after 3 months. 11 out of the 27 T1D patients maintained clinical remission after 1 year. 3 of the MSC transplanted T1D patients became transiently insulin independent. There was also a transient decrease in HbA1c serum levels.	[577]
11 T1D patients received autologous bone marrow-derived MSCs. Early transplantation of MSCs reduced the number of grade II hypoglycemic events. Early MSC transplantation increased the serum levels of IL-4 and increased the proportion of regulatory T cells. These authors found that early MSC transplantation has better clinical outcomes than late MSC transplantation.	[482]
A subgroup of T2D patients that got repeated infusion of umbilical cord-derived MSCs showed reduced HbA1c level and increased glucose infusion rate (GIR), with no overall improvement in islet β-cell function.	[578]
6 T1D children received autologous hematopoietic stem cell transplantation without immunoablation. The MSC transplantation resulted in lower blood glucose levels and HbA1c levels. There was also a decrease in the levels of auto-antibodies against islet cells (ICA), glutamic acid-decarboxylase (GAD), and islet antigen-related tyrosine phosphatase 2 (IA2).	[579]
A Phase I/II study treating recent onset T1D patients with allogeneic Wharton’s jelly MSCs showed that MSCs prevented the drop in serum C-peptide levels and prevented the increase in insulin requirement seen in the placebo group.	[490]

## Data Availability

Raw data are available upon reasonable request.

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
