# Peer review of "A Supportive Role of Mesenchymal Stem Cells on Insulin-Producing Langerhans Islets with a Specific Emphasis on The Secretome"

_biomedicines, 2023, doi:10.3390/biomedicines11092558_

Round 1

Reviewer 1 Report

The Ms by Ronit Vogt Sionov and Ronit Ahdut-HaCohen focused on the contribute of MSCs in pancreatic β cell reconstitution. The authors provided an overview of the mechanisms and secretome associated with MSC-mediated regenerative actions. Th Ms is well written and organized, however, since together with MSCs their derived EVs display regenerative activity, the authors should add  at least a reference in which EV regenerative potential are discussed.

In addition, the contribute of HLSC in type 1 DM must be also reported (PMID: 30191384; PMID: 31025308).

Finally, a clear picture reporting the most relevant mechanisms associated with MSCs in pancreatic β cell reconstitution must be included.

Author Response

Point-to-point response to Reviewer 1

We thank the reviewer for critically reviewing our manuscript and for the good comments that have significantly improved our manuscript.

The Ms by Ronit Vogt Sionov and Ronit Ahdut-HaCohen focused on the contribute of MSCs in pancreatic β cell reconstitution. The authors provided an overview of the mechanisms and secretome associated with MSC-mediated regenerative actions. Th Ms is well written and organized, however, since together with MSCs their derived EVs display regenerative activity, the authors should add  at least a reference in which EV regenerative potential are discussed.

We have accordingly added a section discussing the potential therapeutic applications of MSC-derived extracellular vesicles in regenerative medicine (lines 650-661) as well as text to lines 640-649.

In addition, the contribute of HLSC in type 1 DM must be also reported (PMID: 30191384; PMID: 31025308).

Accordingly, text has been added to lines 450-452, mentioning the generation of insulin-producing 3D spheroid structures from HLSC.

Finally, a clear picture reporting the most relevant mechanisms associated with MSCs in pancreatic β cell reconstitution must be included.

We have now added a figure emphasizing the beneficial effect of MSCs on pancreatic β-cells (Figure 7).

Reviewer 2 Report

Mesenchymal stem cells (MSCs) can support human pancreatic islet function and islet co-transplantation with MSCs is more effective than islet transplantation alone in attenuating diabetes progression. Current review article described various aspects of MSCs related to islet function and diabetes. I like to give the following comments.

1.      Backgrounds and parameters related to pancreatic beta-cells were described in a good way. However, the association of MSCs with these parameters were not conducted. Why?

2.      Role of MSCs in type-1 diabetes (T1DM) has been proposed for a while. However, application of MSCs was not developed markedly. Why?

3.      Source of each figure needs to show in the legends. Materials and methods were required once it was completed in current report.

4.      How to get the Supplementary Materials? It needs to show in clear.

5.      The MSC secretome of diverse factors may influence various aspects of diabetes pathogenesis. This conclusion is better to show a graphic picture in figure.

6.      Patients with T1DM must receive insulin injection to main the life. Effects of exogenous insulin on MSCs were not conducted. Why?

Author Response

Point-to-point response to Reviewer 2

We thank the reviewer for critically reviewing our manuscript and for the comments that have significantly improved our manuscript.

Mesenchymal stem cells (MSCs) can support human pancreatic islet function and islet co-transplantation with MSCs is more effective than islet transplantation alone in attenuating diabetes progression. Current review article described various aspects of MSCs related to islet function and diabetes. I like to give the following comments.

  1. Backgrounds and parameters related to pancreatic beta-cells were described in a good way. However, the association of MSCs with these parameters were not conducted. Why?

This is a review paper that first deals with various aspects of pancreatic β-cells and diabetes, followed by a discussion on the beneficial roles of MSCs in preserving pancreatic β-cells and ameliorating diabetes (Section 4). In Section 4, the various effects of MSCs on parameters related to pancreatic β-cells have been discussed.

  1. Role of MSCs in type-1 diabetes (T1DM) has been proposed for a while. However, application of MSCs was not developed markedly. Why?

Table 4 presents clinical trials that have been conducted for treating diabetic patients with MSCs.

  1. Source of each figure needs to show in the legends. Materials and methods were required once it was completed in current report.

All figures in the review paper are original figures prepared for this manuscript. The data of the figures have not been published previously.  They are based on original studies performed by the authors and related to issues discussed in the review paper. The relevant material and methods for these figures have now been added to Supplementary data.

  1. How to get the Supplementary Materials? It needs to show in clear.

Supplementary Figures appear in the MDPI website below the manuscript (main text).

  1. The MSC secretome of diverse factors may influence various aspects of diabetes pathogenesis. This conclusion is better to show a graphic picture in figure.

We have accordingly prepared a summary figure of the influence of MSCs and its secretome on various aspects of diabetes pathogenesis (Figure 7).

  1. Patients with T1DM must receive insulin injection to main the life. Effects of exogenous insulin on MSCs were not conducted. Why?

As described in Section 4 (lines 427-443), many protocols have been developed to differentiate MSCs into insulin-producing cells, including the addition of insulin. Also, the incubation of MSCs with islets that secrete insulin, may result in insulin-producing MSCs. There is a paper describing increased glucose uptake and GLUT4 translocation upon exposure of MSCs to insulin, that we now have added to the manuscript (lines 446-448).

Reviewer 3 Report

The investigation on the A Supportive Role of Mesenchymal Stem Cells on Insulin-Producing Langerhans Islets with a Specific Emphasis on The Secretome. In my opinion, the work is not good results. In its present form, the manuscript is not fit for publication.

This manuscript review or research? Can authors explain?

Review manuscript means missing copyright of Fig 1,2,3,4,5 and 6.

Research manuscript means missing material methods and ethical or cell line detail 

Moderate editing of English language required

Author Response

Point-to-point response to Reviewer 3

We thank the reviewer for critically reviewing our manuscript and for the comments.

The investigation on the A Supportive Role of Mesenchymal Stem Cells on Insulin-Producing Langerhans Islets with a Specific Emphasis on The Secretome. In my opinion, the work is not good results. In its present form, the manuscript is not fit for publication.

This manuscript review or research? Can authors explain?

This is a review article discussing different aspects of pancreatic β-cells, diabetes and the beneficial role of MSCs on pancreatic β-cells with a specific emphasis on the secretome. The authors have some unpublished raw data cumulated during the years that support and accomplish the data discussed in the review, and in such, have been integrated in the text. These data might find interest for researchers working in the field.

Review manuscript means missing copyright of Fig 1,2,3,4,5 and 6.

These figures are original figures prepared for this review manuscript based on original findings by the authors, and have not been published before.

Research manuscript means missing material methods and ethical or cell line detail 

The relevant material and methods for the figures have now been added to Supplementary data.

Round 2

Reviewer 3 Report

Accept in present form

 Extensive editing of English language required